# Spatial colocalization and molecular crosstalk of myofibroblastic CAFs and tumor cells shape lymph node metastasis in oral squamous cell carcinoma

Ken Furudate[1,2,3,4*], Shuya Kasai[5], Tadashi Yoshizawa[6], Yuya Sasaki[4,7], Kohei Fujikura[8], Shintaro Goto[6], Ryohei Ito[1], Koki Takagi[1], Tomoyuki Tanaka[4], Hiroshi Kijima[6], Kosei Kubota[1], Ken Itoh[5], Wataru Kobayashi[1], Koichi Takahashi[3,4*]

1 Department of Oral and Maxillofacial Surgery, Hirosaki University Graduate School of Medicine, Hirosaki, Japan, 2 Department of Medical Data Intelligence, Research Center for Health-Medical Data Science, Hirosaki University Graduate School of Medicine, Hirosaki, Japan, 3 Department of Genomic Medicine, The University of Texas MD Anderson Cancer Center, Houston, Texas, United States of America, 4 Department of Leukemia, Division of Cancer Medicine, The University of Texas MD Anderson Cancer Center, Houston, Texas, United States of America, 5 Department of Stress Response Science, Biomedical Research Center, Hirosaki University Graduate School of Medicine, Hirosaki, Japan, 6 Department of Anatomic Pathology, Hirosaki University Graduate School of Medicine, Hirosaki, Japan, 7 Department of Hematology, University of Tsukuba Hospital, Tsukuba, Japan, 8 Department of Diagnostic Pathology, Kobe University Graduate School of Medicine, Kobe, Japan

* furudate@hirosaki-u.ac.jp (KF); ktakahashi@mdanderson.org (KT)

## Abstract

Lymph node metastasis (LNM) is a critical prognostic factor for patients with oral squamous cell carcinoma (OSCC). Previous research has implicated the partial epithelial-to-mesenchymal transition of tumor cells and myofibroblastic cancer-associated fibroblasts (myCAFs) in the LNM process. However, the underlying molecular mechanisms remain poorly understood. Here, we conducted a comprehensive molecular analysis integrating original and publicly available OSCC data from bulk genome and transcriptome, single-cell transcriptome, and spatial transcriptome analyses. We found that myCAFs were quantitatively and functionally activated in LNM-positive samples and spatially colocalized with OSCC cells within the invasive tumor front (ITF), providing a niche that may facilitate LNM. Immunohistochemical validation in 90 ITF samples confirmed significantly higher myCAF density in LNM-positive samples than in LNM-negative samples, and this density remained an independent predictor of LNM when adjusted for pathological grade and the pattern of invasion. In LNM-positive samples, myCAFs provided increased extracellular matrix (ECM) signals, upregulating stemness-related genes such as *CD44* in OSCC cells. The functional importance of this myCAF-driven ECM-CD44 axis was further supported by our validation analysis of expanded, publicly available spatial transcriptome and experimental *in vitro* coculture data. We also extracted a spatially resolved, 23-gene signature from the metastatic ITF where OSCC and myCAFs colocalize.

**Data availability statement:** Raw spatial RNA-seq data generated and analyzed during the current study are available in the DNA Data Bank of Japan (RRID:SCR_002359, https://www.ddbj.nig.ac.jp) under accession number PRJDB13905 (https://ddbj.nig.ac.jp/search/entry/bioproject/PRJDB13905). The raw sequence reads are available through the DDBJ Sequence Read Archive (DRA) under the following accession numbers: experiments DRX377812, DRX377813, DRX377814, DRX377815, which correspond to runs DRR391952, DRR391953, DRR391954, DRR391955. Processed spatial transcriptome data can be accessed through the Genomic Expression Archive (GEA, https://ddbj.nig.ac.jp/public/ddbj_database/gea/) under accession number E-GEAD-511 (https://ddbj.nig.ac.jp/public/ddbj_database/gea/experiment/E-GEAD-000/E-GEAD-511/). This GEA deposit (E-GEAD-511) provides a processed AnnData object (.h5ad) integrating data from these four samples, including annotations and clustering results. Supplementary spatial metadata components (pickled Python objects corresponding to AnnData.uns['spatial'] elements), which can be used for specific spatial visualizations with this integrated AnnData object, are available on Figshare (https://doi.org/10.6084/m9.figshare.20408067). Furthermore, the complete 10x Genomics Space Ranger output files for each of the four individual Visium samples are publicly available on Figshare (https://doi.org/10.6084/m9.figshare.29237660). This dataset includes, for each sample: The filtered feature-barcode matrix in HDF5 format (filtered_feature_bc_matrix.h5). The spatial folder containing high-resolution tissue images (tissue_hires_image.png), low-resolution tissue images (tissue_lowres_image.png), fiducial alignment images (aligned_fiducials.jpg), tissue detection images (detected_tissue_image.jpg), scale factors (scalefactors_json.json), and spot coordinates (tissue_positions_list.csv). A patho_annot folder containing annotation files (TXT format) that link pathologist-derived and computational annotations (such as primary pathological classification, graph-based cluster assignments, detailed pathological diagnostic information, broader categories, and descriptive text) to spot barcodes. All patient-derived images within this Figshare dataset were de-identified prior to deposition. Source data files

This signature predicted LNM status and poor overall survival in patients with OSCC. Our findings provide novel insight into the molecular myCAF/OSCC crosstalk that facilitates LNM and identify potential prognostic biomarkers and therapeutic targets for patients with OSCC.

## Author summary

The spread of oral squamous cell carcinoma (OSCC) to lymph nodes leads to poor patient survival. It is therefore essential to understand how OSCC cells invade and interact with the surrounding tissue. Previous studies have shown the involvement of supportive cells, but the exact details and interactions have remained unclear. Using advanced spatial and molecular methods, we found that specific supportive cells called myofibroblastic cancer-associated fibroblasts (myCAFs) are more numerous and active in patients whose OSCC has spread to lymph nodes. These myCAFs gather at the invasive tumor front (ITF), where they interact with OSCC cells directly. At the ITF, myCAFs produce extracellular matrix signals that enhance the invasive and stem-like properties of OSCC cells, partly by increasing expression of the *CD44*. We identified a spatially resolved 23-gene signature at the ITF that predicts lymph node spread and survival outcomes. Our results clarify how interactions between supportive and tumor cells drive OSCC spread and provide potential biomarkers and therapeutic targets.

## Introduction

Oral squamous cell carcinoma (OSCC), which arises from the mucosal epithelium of the oral cavity, represents over 90% of malignant tumors diagnosed in the oral and maxillofacial region [1]. As a locally invasive cancer, OSCC frequently causes lymph node metastasis (LNM) in its early stages, which substantially affects the prognosis of OSCC [2–4]. This metastasis stems from the complex interplay between tumor and stromal cells in the tumor microenvironment (TME) and involves tumor-cell invasion into the extracellular matrix (ECM), the tumor cells' survival in the vascular system, and the formation and proliferation of secondary tumors at the metastatic sites [5,6]. Previous studies of OSCC have suggested an association between LNM pathogenesis and the partial epithelial-to-mesenchymal transition of tumor cells [7–9] or cancer-associated fibroblasts (CAFs) [10,11]. An increased presence of myofibroblastic CAFs (myCAFs) in the TME correlates with unfavorable prognostic factors such as advanced tumor stage and recurrence [12–14]. However, the mechanistic interplay between myCAFs and tumor cells that shapes LNM pathogenesis remains elusive. Furthermore, the lack of diagnostic and prognostic biomarkers related to LNM has impeded improvements in clinical outcomes for patients with OSCC.

To better understand the molecular mechanisms underpinning LNM in OSCC, we conducted a comprehensive molecular analysis integrating both original and publicly

underlying the figures presented in this study, including numerical data for plots, statistical summaries, and other files as described in the repository, are available on Figshare (https://doi.org/10.6084/m9.figshare.21152584). These files are organized by figure number to facilitate reproduction of the presented results. All patient-derived images or data within this specific deposit have been appropriately de-identified. The trained model used in this study is also publicly accessible and can be found on Figshare (https://doi.org/10.6084/m9.figshare.20279025.v1). The source code can be found at https://kenflab.github.io/oscc_metastasis/. The bulk OSCC data are available in TCGA (head and neck squamous cell carcinoma [HNSCC]-TCGA, research resource identifier [RRID]:SCR_003193, https://www.cancer.gov/about-nci/organization/ccg/research/structural-genomics/tcga). The curated TCGA survival datasets for HNSCC are available from the University of California, Santa Cruz's Xena platform (RRID:SCR_018938, https://xenabrowser.net/datapages/). The OSCC microarray datasets were based on the Affymetrix human genome U133 Plus 2.0 Array (accession numbers GSE41613 and GSE42743) and associated survival data from the Gene Expression Omnibus (GEO) database (RRID:SCR_005012, https://www.ncbi.nlm.nih.gov/geo/). The single-cell OSCC transcriptome data used in this study are available from the GEO database under accession number GSE103322. The 12 additional transcriptomic OSCC samples are available from the GEO database under accession number GSE208253 and in vitro co-culture datasets are available from the GEO database under accession number GSE279481 and GSE178153/GSE178154.

**Funding:** This research was partially funded by the Japan Society for the Promotion of Science KAKENHI Program (https://www.jsps.go.jp/english/e-grants/, grant numbers JP 17K17233 and JP 25K12984 [to K.F.] and JP 20K18716 and JP 24K13104 [to R.I.]) and by the Uehara Memorial Foundation grant (https://www.taisho.co.jp/global/sustainability/social/society/uehara_foundation.html, to K.F. and Y.S.). The funders had no role in study design, data collection and analysis, decision to publish, or preparation of the manuscript.

**Competing interests:** The authors have declared that no competing interests exist.

available OSCC data from bulk genome and transcriptome analyses, single-cell transcriptome profiling, and spatial transcriptome analyses. We found that myCAFs are quantitatively and functionally activated in LNM-positive samples and are spatially colocalized with OSCC cells within the invasive tumor front (ITF). Mechanistically, myCAFs provided increased ECM signals to OSCC cells via genes such as the genes in the collagen gene family, fibronectin 1 (*FN1*), and the genes in the laminin gene family (*LAMA*s, *LAMB*s). Consequently, OSCC cells upregulated ECM receptors (e.g., α integrins [*ITGA*s], β integrins [*ITGB*s], syndecans [*SDC*s], and *CD44*), which led to upregulation of stemness-related genes in OSCC cells. Furthermore, we extracted a spatially resolved, 23-gene signature that predicted LNM status and poor overall survival (OS) in patients with OSCC. These findings provide novel insights into the role of myCAFs in the TME and offer clinically valuable molecular biomarkers for patients with OSCC.

## Results

### Bulk transcriptome data revealed upregulated expression of muscle contraction genes in LNM-positive samples

To identify genomic and transcriptomic determinants of LNM in patients with OSCC, we first analyzed whole-exome sequencing (WES) and RNA sequencing (RNA-seq) data from OSCC patients studied in The Cancer Genome Atlas (TCGA) project. For this study, we excluded human papilloma virus (HPV)-positive cases, as described in the Methods and Discussion. This resulted in the analysis of a total of 201 HPV-negative OSCC cases:115 patients with LNM (tumor-node-metastasis [TNM] stage, N1-N3) and 86 patients without LNM (TNM stage, N0; Figs 1A and S1A–S1B and S1 Table). The most commonly detected cancer-driver mutations were tumor protein p53 (*TP53*) mutations (74%; *n*=148), followed by mutations in titin (*TTN*; 32%, *n*=64), FAT atypical cadherin 1 (*FAT1*; 23%; *n*=47), cyclin-dependent kinase inhibitor 2A (*CDKN2A*; 21%; *n*=42), notch receptor 1 (*NOTCH1*; 17%; *n*=34), phosphatidylinositol-4,5-bisphosphate 3-kinase catalytic subunit alpha (*PIK3CA*; 17%; *n*=34), mucin 16 (*MUC16*; 15%; *n*=30), spectrin repeat-containing nuclear envelope protein 1 (*SYNE1*; 15%; *n*=30), and caspase 8 (*CASP8*; 14%; *n*=28; Fig 2A). When we analyzed the correlation between LNM status and the driver mutation profiles, none of the driver mutations showed a significant association with LNM status (Fig 2B and S2 Table).

We then analyzed RNA-seq data to identify the differentially expressed genes (DEGs) associated with LNM status and found 260 significantly upregulated DEGs (Fig 2C). Pathway analysis of these DEGs revealed that the genes involved in muscle contraction, including actin gamma 2 (*ACTG2*), ATPase sarcoplasmic/endoplasmic reticulum Ca2+ transporting 1 (*ATP2A1*), myosin heavy chain 6 (*MYH6*), myosin heavy chain 8 (*MYH8*), myosin heavy chain 11 (*MYH11*), myosin light chain 3 (*MYL3*), myosin light chain 4 (*MYL4*), and myosin light chain 7 (*MYL7*), were the top significant genes associated with LNM-positive status (Fig 2D). Since the enrichment of muscle contraction genes could have been driven by differences in the tumor purity of the biopsied samples, we used the Estimation of Stromal and Immune Cells in Malignant Tumor Tissues Using Expression Data (ESTIMATE) tool (The University

PLOS Genetics

## A

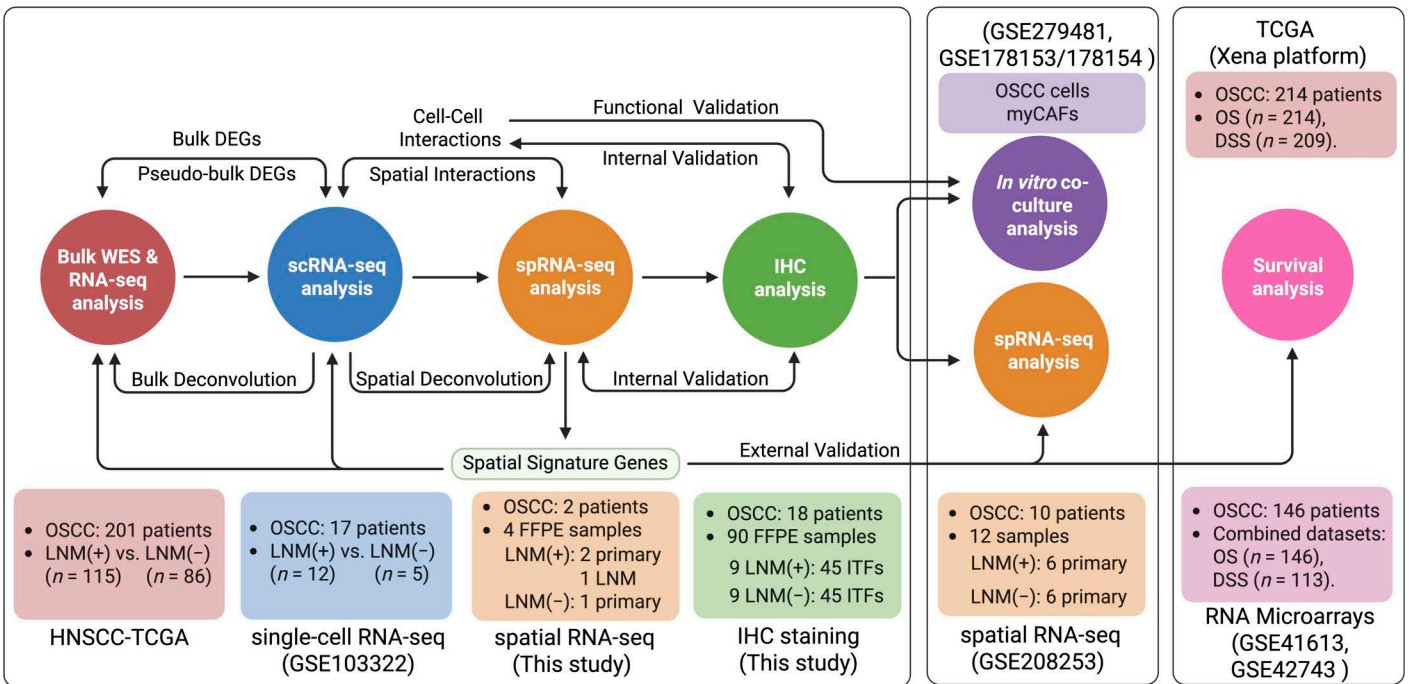

**Fig 1. Schematic overview of multimodal OSCC data integration.** (A) Schematic overview of this study, which integrated multimodal human oral squamous cell carcinoma (OSCC) data from bulk whole-exome sequencing (WES), RNA sequencing (RNA-seq), microarray, single-cell RNA sequencing (scRNA-seq), spatial RNA sequencing-(spRNA-seq), and immunohistochemical (IHC) analyses. Created in BioRender.

of Texas MD Anderson Cancer Center) to deconvolute RNA-seq data and infer the tumor purity of the samples [15]. When comparing the primary tumor samples from patients with or without LNM, we found no significant differences in the abundance of tumor stromal cells ($P = 0.95$; 95% CI, −205.59-249.37) or immune cells ($P = 0.87$; 95% CI, −247.47-203.15; S1C Fig), suggesting that the observed DEGs in the LNM-positive samples were not caused by differences in sampling.

### Single-cell transcriptome data revealed a high abundance of and enhanced intercellular communication between myCAFs in the LNM-positive OSCC microenvironment

To better understand the cellular origin of the observed DEGs in the bulk RNA-seq data, we then analyzed the publicly available single-cell RNA-seq (scRNA-seq) data of patients with OSCC (source: GSE103322 [7]). These data included a total of 5,884 cells from 17 patients with OSCC (12 LNM-positive patients and 5 LNM-negative patients; Figs 1A and S2A–S2D and S3 Table).

We first compared the DEGs of pseudo-bulk cells from the primary tumor site using the samples from the 17 patients with OSCC. Consistent with the bulk RNA-seq results from TCGA (Fig 2D), the DEGs highly expressed in the cells from the LNM samples were involved in pathways related to muscle contraction (S2E Fig). Next, we mapped the scRNA-seq data using the uniform manifold approximation and projection method and subsequently identified clusters of patients with OSCC. For each patient with LNM, the primary tumor and metastatic sites were clustered together, suggesting that the transcriptomic status of tumor cells between the primary tumor and metastatic sites was similar (Figs 3A–3C and S2F–S2H). We then examined the expression of the muscle contraction–related genes for each cell type, and the myCAFs

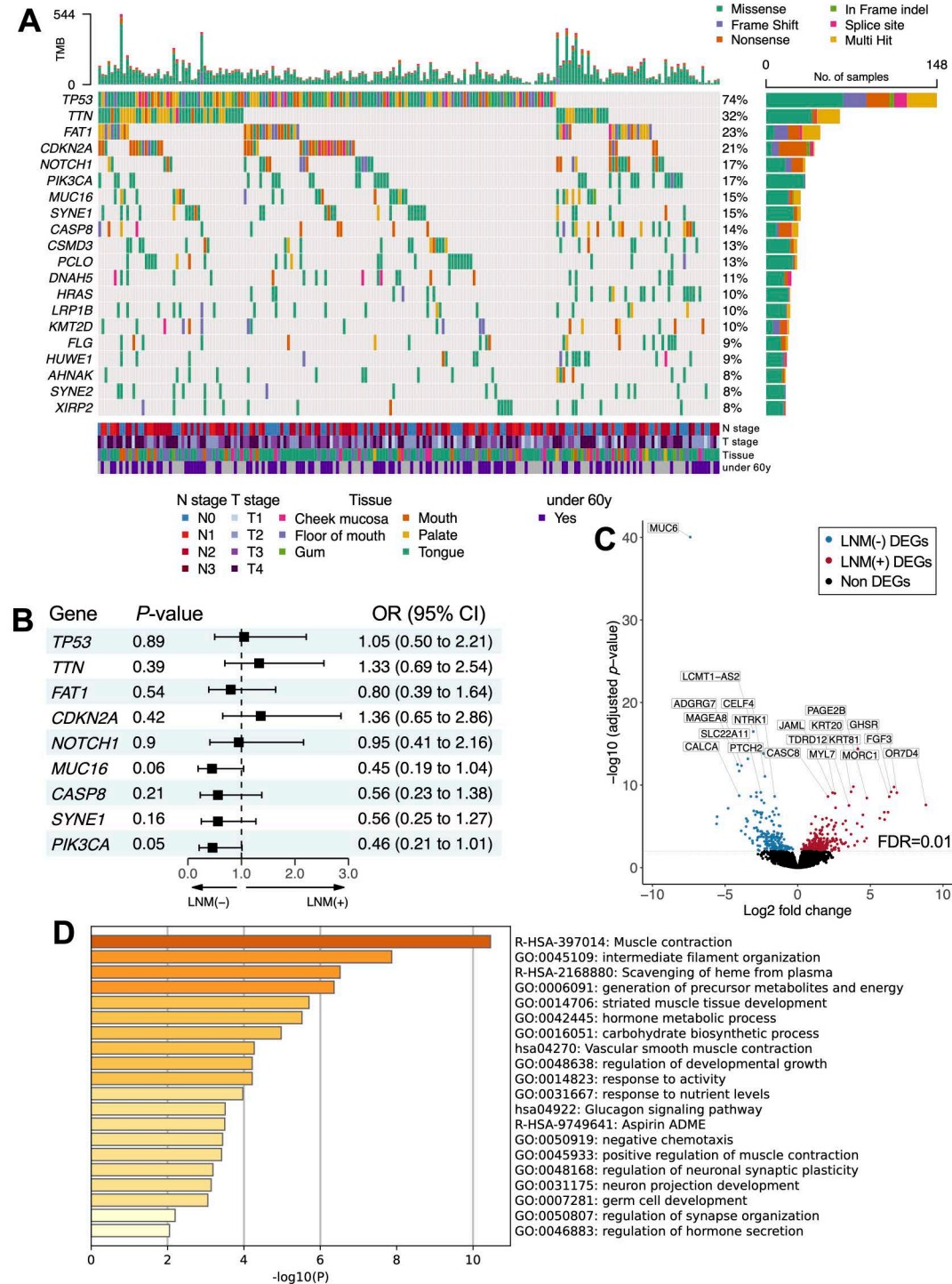

**Fig 2. Bulk analysis of metastasis-related factors in OSCC.** (A) Oncoplot displays the 20 most frequently mutated genes in 201 patients with oral squamous cell carcinoma (OSCC) and is based on TCGA-head and neck squamous cell carcinoma (HNSCC) WES data. Rows denote genes arranged by mutation frequency. Gene names are on the left and mutation frequencies are on the right. Columns represent individual patients, and a color bar beneath the oncoplot provides patients' clinical details, including the tumor-node-metastasis (TNM) stage (lymph node metastasis [LNM] status is indicated as an N stage, and tumor size is indicated as a T stage), primary tissue, and age categorization (under 60 years [y]). To the right, stacked bars show mutation tallies, colored by mutation type, for each gene. A key above these bars identifies the mutation types. Within the main plot, rectangles signify diverse somatic mutations. The bars at the top reflect the tumor mutation burden (TMB) of each patient, with the colors corresponding to the

mutation types in the heatmap. (B) Forest plot depicts the associations between the 9 most frequently mutated genes (*TP53, TTN, FAT1, CDKN2A, NOTCH1, MUC16, CASP8, SYNE1,* and *PIK3CA*) and LNM status among 201 patients with OSCC. Squares represent odds ratios (ORs), and bars display 95% confidence intervals (CIs). ORs were resolved from a multiple logistic regression analysis. (C) Volcano plot shows 260 upregulated differentially expressed genes (DEGs) in 201 patients (LNM-positive or LNM-negative) with OSCC. The genes were identified using Tag Count Comparison (TCC) with iDEGES/edgeR normalization and the Fisher exact test (false discovery rate [FDR] < 0.01). LNM-positive DEGs are red, LNM-negative DEGs are blue, and the top DEGs are labeled. (D) Heatmap shows the top enriched pathways from the Metascape analysis of the DEGs associated with LNM status. Each row represents a pathway. The heatmap illustrates the top nonredundant enrichment pathways using a discrete color scale to indicate statistical significance. The pathway R-HSA-397014, the muscle contraction pathway, was the most enriched pathway in the LNM-positive patients. Abbreviations: *CASP8*, caspase 8; *CDKN2A*, cyclin-dependent kinase inhibitor 2A; *FAT1*, FAT atypical cadherin 1; FFPE, formalin-fixed, paraffin-embedded; ITF, invasive tumor front; *MUC16*, mucin 16; *NOTCH1*, notch receptor 1; OS, overall survival; *PIK3CA*, phosphatidylinositol-4,5-bisphosphate 3-kinase catalytic subunit alpha; *SYNE1*, spectrin repeat-containing nuclear envelope protein 1; TCGA, The Cancer Genome Atlas *TP53*, tumor protein p53; *TTN*, titin.

showed the highest expression of muscle contraction genes (Fig 3D). Although we did not find a statistically significant difference in the relative fractions of each cell type between LNM-positive and LNM-negative samples, there was a numerically higher fraction of myCAFs in patients with LNM compared with those without (median fractions: 13% [interquartile range, 5%-24%] vs. 4% [interquartile range, 1%-16%], respectively; *P* = 0.15; Fig 3E). The lack of statistical significance was likely due to the small sample size. These results were further corroborated with the bulk TCGA RNA-seq data, in which computational deconvolution using CIBERSORTx revealed significant enrichment of myCAFs in patients with LNM progression (*P* = 0.049; Spearman correlation = 0.13; Figs 3F–3G and S2I). These results suggest that a higher abundance of myCAFs is associated with LNM in OSCC.

Next, to assess the functional activity of stromal cells—including myCAFs—in OSCCs, we used CellChat [16], which generates *in silico* inferences for cell-cell interactions using scRNA-seq data. Overall, the LNM-positive samples showed enhanced cell-cell interactions compared with the LNM-negative samples (number of inferred interactions: 4,902 [95% CI: 3,616-5,001] vs. 2,731 [95% CI: 2,004-3,529]). Moreover, the strength of the intercellular interactions was greater in the LNM-positive samples compared with the LNM-negative samples (interaction strength: 120.37 [95% CI: 96.63-146.83] vs. 88.07 [95% CI: 58.48-123.13]; Fig 4A). Among the cell types, the myCAFs exhibited the highest differential outgoing interaction when the LNM-positive and LNM-negative samples were compared (Figs 4B and S2J). Specifically, we observed that the myCAFs expressed higher levels of ECM-related genes, including those in the collagen gene family (collagen type 1 alpha 1 chain [*COL1A1*], collagen type 1 alpha 2 chain [*COL1A2*], collagen type IV alpha 1 chain [*COL4A1*], collagen type IV alpha 2 chain [*COL4A2*], collagen type VI alpha 1 chain [*COL6A1*], and collagen type VI alpha 2 chain [*COL6A2*]), *FN1*, and the laminin gene family [*LAMA5, LAMB1, LAMB2*], in LNM-positive samples (S2K Fig). We also found that OSCC cells upregulated ECM-associated receptors, specifically *ITGAs, ITGBs, SDCs,* and *CD44* (Figs 4C and S2L). CD44 is a well-known marker for cancer stem cells [17] and has been associated with poor prognosis and treatment resistance in OSCC [18–20]. We found that the OSCC cells' cancer stemness scores [21,22] were significantly higher in LNM-positive samples than in LNM-negative samples (*P* < 0.001 and 95% CI, 0.06-0.1 vs. *P* < 0.001 and 95% CI, 0.02-0.05, respectively; S2M–S2N Fig). Further analysis using CRISPRGeneEffect data from DepMap [23] found that *ITGB1, ITGA3,* and *CD44* were among the most essential genes for the survival and growth of OSCC cells (S2O Fig). Taken together, these findings suggest that myCAFs were not only abundant in LNM-positive samples but also functionally activated and providing ECM-related signals to OSCC cells. The ECM-related signals may have subsequently activated stemness-related genes such as *CD44* in OSCC cells.

### Spatial transcriptomic analysis revealed colocalization of myCAFs and OSCC cells at the ITF

Next, to investigate how cellular interactions between OSCC cells and myCAFs are spatially organized, we performed a spatial transcriptomic analysis of 4 surgical specimens collected from 2 patients with OSCC treated at Hirosaki University Hospital. The clinical characteristics of these 2 patients (HUH001 and HUH002) are described in the S1 Methods and S4

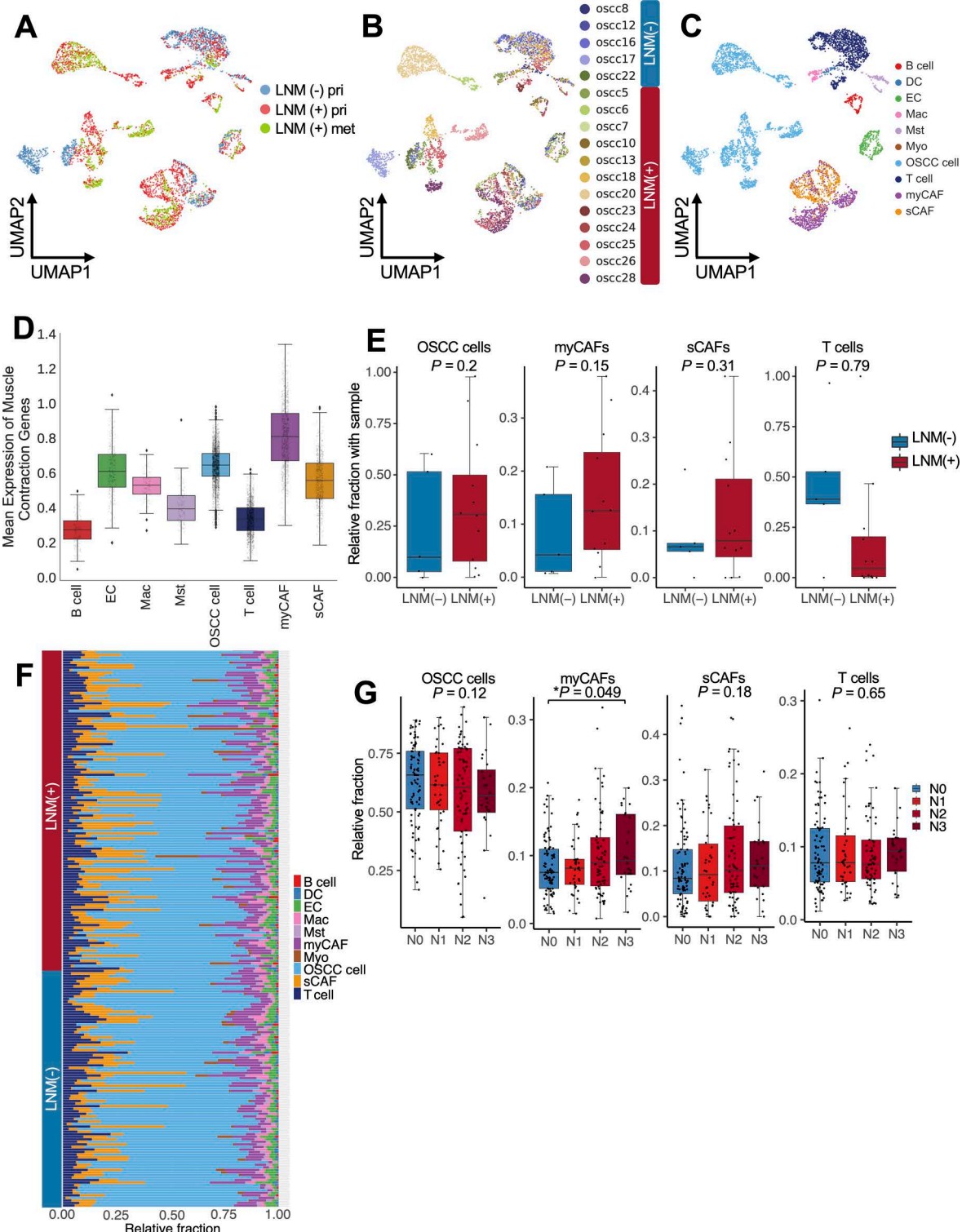

**Fig 3. Enhanced intercellular interaction of myCAFs within the OSCC microenvironment in LNM.** (A-C) Uniform manifold approximation and projection (UMAP) visualizations of 5,884 cells from 17 oral squamous cell carcinoma (OSCC) samples analyzed using single-cell RNA sequencing. (A) UMAP visualization of primary (pri) tumor sites without lymph node metastasis (LNM) (blue), primary tumor sites with LNM (red), and metastatic (met) sites (green). (B) UMAP visualization of cells using sample identifier numbers. (C) UMAP visualization of cells grouped into 10 key cell types: B cells,

dendritic cells (DCs), endothelial cells (ECs), macrophages (Macs), mast cells (Msts), myocytes (Myos), OSCC cells, T cells, myofibroblastic cancer-associated fibroblasts (myCAFs), and secretory/matrix-remodeling cancer-associated fibroblasts (sCAFs). See S2F–S2H Fig for further details. (D) Box plots illustrate the average expression of 211 muscle contraction–related genes across 8 distinct cell types, excluding those representing less than 1% of the total cell population. The genes were sourced from the REACTOME_MUSCLE_CONTRACTION database, v2023.1. The y-axis displays the mean expression levels of these genes. Black circles indicate individual cells. The center lines represent the medians, the box borders represent the interquartile ranges (IQRs), and the whiskers represent ±1.5×the interquartile ranges (IQRs). (E) Box plots show the relative fractions of OSCC cells, myCAFs, sCAFs, and T cells within samples, distinguished between cases with LNM (red) and without LNM (blue). Black circles represent individual samples. No statistical significance ($P<0.05$) was observed in any comparison, as determined using a 2-sided Mann-Whitney $U$ test. (F) Relative cellular fractions for 10 cell types in each sample from 201 patients with OSCC. Cellular fractions were estimated via a CIBERSORTx analysis. (G) Relative fractions of OSCC cells, myCAFs, sCAFs, and T cells within each sample were compared across different LNM stages. Significance (indicated with an asterisk) was determined using a Spearman correlation and was observed for the myCAFs ($P=0.049$).

Table. Patient HUH001 had LNM (stage T2N3bM0), whereas patient HUH002 did not (stage T2N0M0). Two primary tissue samples (HUH001-P1 and HUH001-P2) and 1 metastatic tissue sample (HUH001-met) from patient HUH001 and 1 primary tissue sample from patient HUH002 (HUH002-P) were analyzed using the Visium formalin-fixed, paraffin-embedded (FFPE) Spatial Gene Expression platform (Figs 5A,5B and S3A–S3G and S5 and S6 Tables).

To investigate the spatial localization of OSCC cells and myCAFs, we deconvoluted cells within each spot using Tangram [24]. This approach maps single cells to lower-resolution spatial transcriptomic profiles using marker genes identified via scRNA-seq data (S4A–S4E Fig). Qualitatively, in the LNM-positive samples (HUH001-P1 and HUH001-P2), the myCAFs spatially colocalized with the OSCC cells at the ITF, which is the pathologically defined edge of a malignant tumor where cancer cells are actively invading into the surrounding normal tissue (Figs 6A,S3A,S4A and S4B). Conversely, in the LNM-negative sample (HUH002-P), the myCAFs spatially segregated from the OSCC cells in the primary tumor site, and only OSCC cells were present within the ITF (Figs 6A and S4D). Quantitatively, the co-localization of the OSCC cells and myCAFs was significantly higher in the HUH001 (LNM-positive) specimens than in the HUH002 (LNM-negative) specimens ($P=0.003$; odds ratio, 2.17; 95% CI, 1.40-3.38; Fig 6C–6D). Immunohistochemical (IHC) analysis of the corresponding FFPE samples also confirmed this spatial co-localization of myCAFs and OSCC cells (Fig 6B).

To validate these findings in a larger cohort, we then conducted an IHC analysis of 90 additional samples of ITFs from 18 patients with OSCC (9 with LNM and 9 without LNM; Fig 7A). All selected patients had stage T2 primary tumors as determined by the tumor sizes and locations (S4 Table). Consistent with the findings from the spatial transcriptomic analysis, significantly higher absolute α-smooth muscle actin (α-SMA)–positive areas per square millimeter (myCAF densities) were detected in the ITFs of samples from patients with LNM compared with those of samples from patients without LNM (median: 19.14% [95% CI, 15.92%-21.99%] vs. 10.70% [95% CI, 9.06%-13.12%], respectively; $P<0.001$; Fig 7B). Moreover, histopathological analyses revealed that the myCAF density did not significantly correlate with the pathological grade or the pattern of invasion (POI; Spearman correlation = −0.16, $P=0.14$ and Spearman correlation = 0.06, $P=0.60$, respectively; S4F Fig). Importantly, in the multivariable logistic regression adjusted for the pathological grade and the POI, myCAF density was identified as an independent predictor of LNM (odds ratio, 1.45; 95% CI, 1.21–1.72; $P<0.001$; Fig 7C), underscoring the association between myCAF presence within the ITF and LNM risk.

Next, we examined the ligand-receptor pairs (myCAFs and OSCC cells) in the spatial transcriptome data using the scRNA-seq data described above. The LNM-positive samples (HUH001-P1 and HUH001-P2) showed enhanced spatial gene expression of inferred receptors such as *ITGAs*, *ITGBs*, *CD44*, and *SDCs* in OSCC cells within the ITF (S4G–S4I Fig). A further spatial interaction analysis confirmed that these LNM-positive samples exhibited increased spatially organized communication compared with the LNM-negative sample (HUH002-P); the identified signaling pathways involved the collagen gene family, *CD44*, and transforming growth factor β (*TGFβ*; S4J Fig).

To validate the proposed myCAF-OSCC cell interactions via the ECM-CD44 axis, we analyzed 12 additional, publicly available, spatial transcriptomic OSCC samples (source: GSE208253 [25]) and *in vitro* co-culture datasets (sources: GSE279481 [26] and GSE178153/GSE178154 [27]). The validation spatial cohort consistently showed a spatially

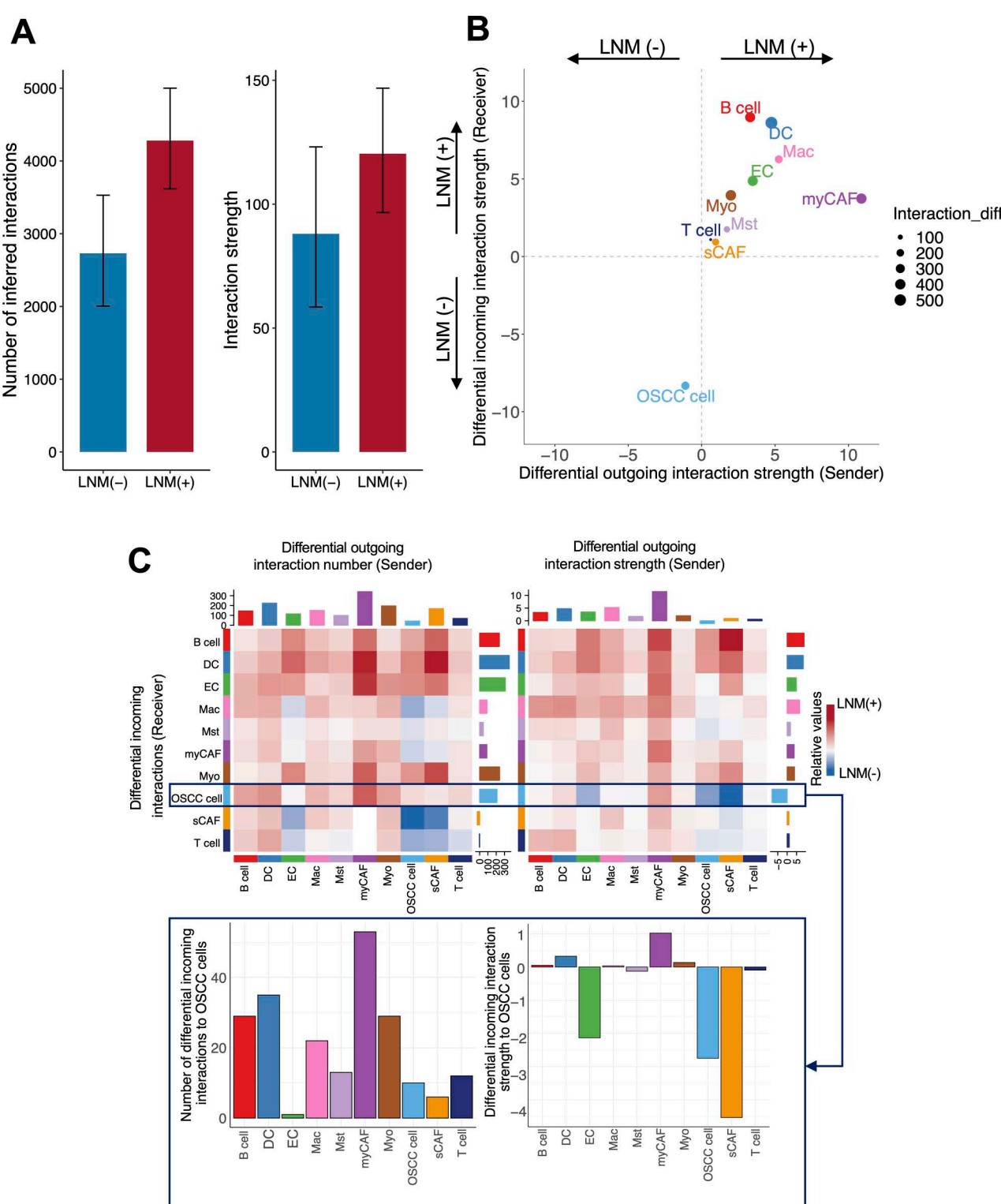

**Fig 4. Differential intercellular communication in OSCC by LNM status.** (A) Bar plots showing the total number (left panel) and overall strength (right panel) of inferred intercellular interactions in lymph node metastasis (LNM)-positive and LNM-negative OSCC samples. Data are presented with 95% boot-strap CIs derived from 10,000 resamples. (B) Scatter plot comparing interaction strengths for each cell type in LNM-positive vs. LNM-negative samples. Outgoing interactions are shown on the x-axis, and incoming interactions are shown on the y-axis. (C) Heatmaps comparing intercellular

communications in LNM-positive vs. LNM-negative samples for each cell type. The bars at the top illustrate outgoing interactions, while the bars on the left display incoming interactions for each cell type. Red indicates LNM-positive interactions, blue indicates LNM-negative interactions, and the color intensity indicates the number and strength of the inferred interactions. The communications received by OSCC cells are depicted below the heatmaps.

organized ECM-CD44 axis. In the LNM-positive samples, this reflected active collagen signaling from myCAFs to OSCCs within the ITFs (S4K–S4N Fig). This spatial organization occurred alongside enhanced myCAF-OSCC colocalization ($P = 0.01$) and significantly higher spatial *CD44* expression in LNM-positive samples compared with LNM-negative samples ($P = 0.02$; S4O–S4P Fig). Analyses of the GSE279481 co-culture dataset provided further support for these findings. A direct comparison of *CD44* expression in OSCC cells cultured with or without myCAFs revealed a numerically higher median expression in OSCC cells co-cultured with myCAFs compared to OSCC cells cultured alone, although this difference was not statistically significant (median: 7.48 [interquartile range, 6.62-8.28] vs. 5.46 [interquartile range, 5.25-6.87], respectively; $P = 0.40$; S4Q Fig). However, hierarchical Bayesian modeling provided evidence supporting a global positive association between myCAF-derived ECM genes and *CD44* expression in OSCC cells (posterior slope mean = 0.27; 95% highest density interval, −0.1-0.63; S4R Fig). Mechanistic modeling of this data further indicated significant indirect effects of myCAF-ECM signaling on *CD44* expression in OSCC cells, mediated by *ITGAs*/*ITGBs* (indirect effect $\beta = 0.29$; 95% CI, 0.17-0.40) and *SDCs* (indirect effect $\beta = 0.47$; 95% CI, 0.34-0.57; S4S Fig). Finally, in an independent microarray co-culture dataset (GSE178153/GSE178154), patient-derived head and neck squamous cell carcinoma (HNSCC) cell lines from LNM-positive tumors showed significantly higher *CD44* expression than those from LNM-negative tumors ($P = 0.04$; S4T Fig). These findings from multiple independent datasets confirmed the intercellular interaction data from the scRNA-seq analysis and provided evidence that these interactions are driven by the spatial colocalization of myCAFs and OSCC cells within the ITF.

### Spatial transcriptomic analysis of matched primary and metastasis samples revealed a transcriptional program shared between primary and metastatic sites

By analyzing the spatial transcriptomic data of the matched primary tumor sites (HUH001-P1 and HUH001-P2) and metastatic site (HUH001-met) from the patient with LNM, we sought to identify the origins of the metastatic cells in the primary tumor site. We used single-cell Variational Inference (scVI; scvi-tools v0.14.5) [28], which is based on probabilistic models and deep neural networks, to integrate the spatial transcriptome data and perform unsupervised clustering. This resulted in 12 clusters that were subsequently restored to their spatial coordinates in the primary tumor (samples HUH001-P1 and HUH001-P2) and metastatic (HUH001-met) sites (Figs 8A, S5A and S5B). Additionally, copy number variations (CNVs) data [29] were used to confirm the origin of metastatic cells in the primary tumor sites at the genomic level (S5C Fig).

We identified 2 main clusters: the clusters predominant in the primary tumor sites (the primary tumor cluster, consisting of clusters 3 and 12) and the clusters shared between the primary tumor sites and the metastatic site (the metastatic cluster, consisting of clusters 1, 2, and 4–11; Figs 8B,8C, S5D and S5E). To explore tumor clusters associated with LNM, we conducted a pseudotime trajectory analysis using stLearn (S5F–S5G Fig) [30]. The metastatic clusters 1, 5, 7, and 9 exhibited shorter pseudotimes than did primary tumor cluster 12 and had elevated *CD44* expression indicating enhanced stemness (all adjusted $P < 0.001$; S10 Table and S5H Fig). DEG analysis revealed unique molecular signatures for these spatial clusters. Cluster 1 showed enrichment of the myCAF-like profile, including desmin (*DES*), troponin I1 (*TNNI1*), and collagen type XI alpha 1 chain (*COL11A1*). Cluster 5 was characterized by elevated serpin family B member 3 (*SERPINB3*), S100 calcium binding protein A9 (*S100A9*), and mesothelin (*MSLN*) expression, indicating highly invasive tumor cells. Cluster 7 demonstrated high expression of markers for class-switched immunoglobulins, notably immunoglobulin heavy constant gamma 4 (*IGHG4*), immunoglobulin heavy constant alpha 1 (*IGHA1*), and immunoglobulin lambda variable 3–1 (*IGLV3-1*). Cluster 9 was distinguished by markers typically associated with CXCL14-positive basal-like epithelial

# A

## Spatial transcriptomic analysis

**1** Spatial Deconvolution and Mapping Analysis using primary tumor samples

**2** Validation of Spatial Findings via IHC staining on 90 additional samples

**3** Expanded Validation by Spatial Data and *In vitro* Co-culture Data

**4** Integrated Analysis of Matched LNM

**5** Extract of 23-gene signature and Validation with External Cohorts

- Using Tangram, mCAFs spatially co-localized with OSCC cells in the ITFs of LNM(+).
- IHC of corresponding FFPE samples confirmed the co-localization of OSCC cells and myCAFs.

- Validated spatial deconvolution findings using IHC analysis on 90 additional ITF samples.
- Using Squidpy, active spatial interactions among the *Collagen, CD44, and TGFβ* in the ITFs of LNM(+).

- Spatial data from 12 additional OSCC samples further validated the spatial co-localization of mCAFs with OSCC cells in the ITFs.
- Functional role of the ECM-CD44 axis was validated by *in vitro* co-culture data.

- Integrated spatial transcriptomic data of primary and metastatic sites from the same patient using scVI.
- From integrated spatial transcriptomic data: 298 upregulated DEGs identified in the Metastatic Cluster.

298 upregulated DEGs in Metastatic Cluster

↓

LASSO-penalized Cox proportional hazards model

↓

23-gene signature

- Extracted a spatially derived 23-gene signature
- Validated the prognostic significance of the 23-gene signature in two independent datasets (GSE41613, GSE42743)

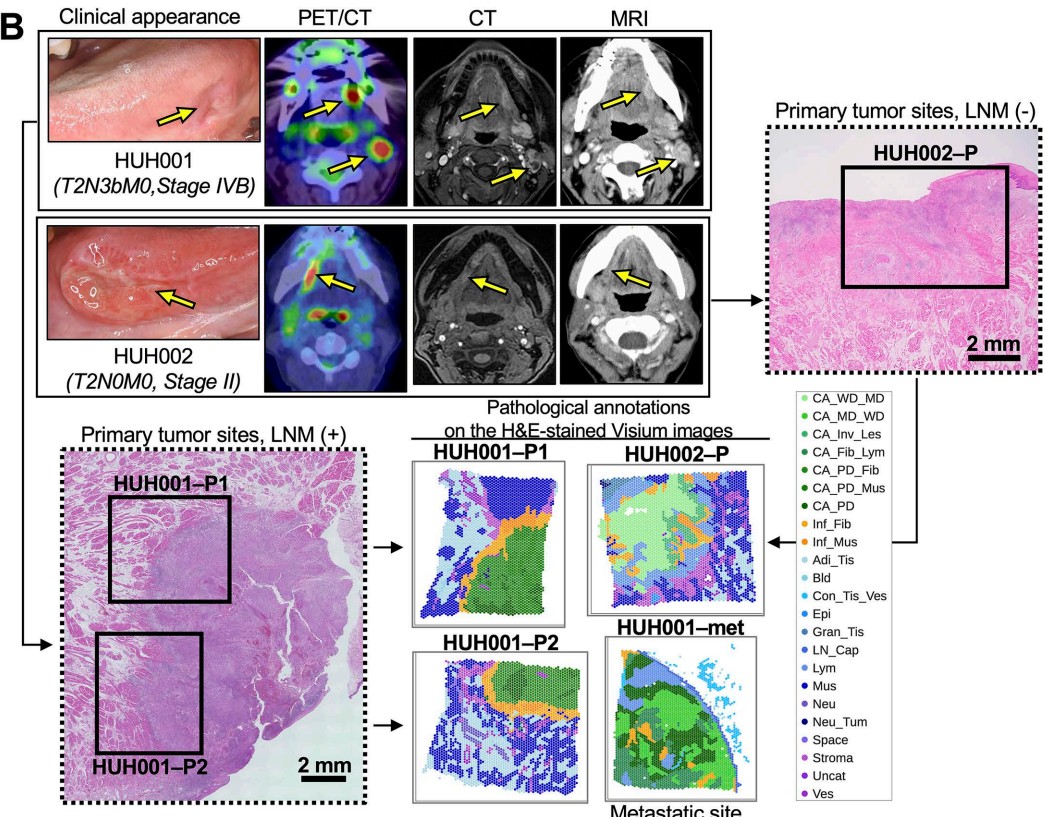

**Fig 5. Exploration of OSCC microenvironment heterogeneity with a spatial transcriptomic analysis.** (A) Schematic overview of the spatial transcriptomic analysis. Created in BioRender. (B) Illustration presenting the clinical appearance and various imaging modalities used for patients with oral squamous cell carcinoma (OSCC), including positron emission tomography/computed tomography (PET/CT), CT, and magnetic resonance imaging (MRI). Four FFPE tissue slides from 2 patients with OSCC, patients HUH001 and HUH002, were analyzed using the Visium formalin-fixed, paraffin-embedded (FFPE) Spatial Gene Expression platform. The 4 slides included 2 primary tissues (HUH001-P1 and HUH001-P2) and 1 metastatic tissue (HUH001-met) from patient HUH001, who had LNM. Additionally, 1 primary tissue (HUH002-P) from patient HUH002, who did not have LNM, was included. Specialist histopathological annotations identifying the invasive tumor fronts (ITFs) in HUH001-P1 and HUH001-P2 were provided. The

PLOS Genetics

adjacent sections from the same specimen were used for diagnostic hematoxylin and eosin/ immunohistochemical images and Visium images, respectively (See also the FFPE sample preparation and hematoxylin and eosin—stained images for spatial transcriptome analysis subsection in the Methods). Abbreviations: CA_WD_MD, cancer, well-differentiated/moderately differentiated; CA_MD_WD, cancer, moderately differentiated/well-differentiated; CA_Inv_Les, cancer, invasive lesions; CA_Fib_Lym, cancer, fibrosis, lymphocytes; CA_PD_Fib, cancer, poorly differentiated fibrosis; CA_PD_Mus, cancer, poorly differentiated muscle; CA_PD, cancer, poorly differentiated; Inf_Fib, inflammation, fibrosis; Inf_Mus, inflammation, muscle; Adi_Tis, adipose tissue; Bld, blood; Con_Tis_Ves, connective tissue, vessels; Epi, epithelium; Gran_Tis, granulation, tissue; LN_Cap, lymph node capillaries; Lym, lymphocytes; Mus, muscle; Neu, neurons; Neu_Tum, neuron tumor; Uncat, uncategorized; Ves, vessels. The depicted histopathological annotations are broadly categorized into shades of green, representing tumor regions; shades of yellow, indicating peritumor regions; and shades of blue, denoting nontumor regions. A 2-mm scale bar is included for reference. For additional information, please refer to S3 Fig.

cells, including C-X-C motif chemokine ligand 14 (*CXCL14*), serine peptidase inhibitor kazal type 6 (*SPINK6*), and keratin 1 (*KRT1*). Interestingly, cluster 1, which had the highest proportion of cells from primary tumor sites among the metastatic clusters, showed enrichment of myCAFs and activated *COL11A1*, suggesting molecular myCAF/OSCC crosstalk though the ECM-CD44 axis. These data depict a continuum of spatial states in the ITF and reflect the migratory, inflammatory, and stem-like characteristics potentially underpinning LNM in OSCC (refer to S11 Table for a full list of cluster characteristics).

To understand the gene-expression profiles unique to metastatic cells, we compared the gene-expression profiles in the 2 main clusters (the primary tumor cluster and the metastatic cluster) and identified 298 significantly upregulated DEGs, including members of the collagen gene family (*COL1A2* and collagen type III alpha 1 chain [*COL3A1*]), decorin (*DCN*), tropomyosin 2 (*TPM2*), immunoglobulin kappa constant (*IGKC*), procollagen C-endopeptidase enhancer (*PCOLCE*), paired related homeobox 1 (*PRRX1*), actin alpha 2 (*ACTA2*), and ring finger protein 213 (*RNF213*; Figs 8D and S5I). The pathway analysis of these DEGs showed significant enrichment of the genes involved in hallmark epithelial-to-mesenchymal transition and hallmark myogenesis (Figs 8E and S5J). Consistent with this finding, these top DEGs showed the highest expression in myCAFs per scRNA-seq data (Fig 8F) and within ITFs of LNM-positive samples from the validation spatial cohort (*P* < 0.001; S5K Fig). Furthermore, we found a significant correlation between the colocalized regions with myCAFs and OSCC cells and those of the ITFs in the metastatic cluster (*P* < 0.001; odds ratio, 2.24; 95% CI, 1.78-2.81; Fig 8G).

## Spatially resolved metastatic cluster genes predicted the clinical outcomes of patients with OSCC

We hypothesized that the 298 DEGs that were found in the spatial analysis and were unique to the metastatic cluster could function as predictive biomarkers for LNM progression and the overall prognosis of patients with OSCC. To test this hypothesis, we first developed a LASSO-penalized Cox proportional hazards model using clinical data and RNA-seq data from TCGA [31,32]. Using this model, we developed a gene signature–based risk score. This approach helped us to reduce our list of DEGs of interest to 23 core signature genes (Fig 9A and S12 and S13 Tables). This 23-gene signature stratified OS and disease-specific survival (DSS) well in the cohort from TCGA (*P* < 0.001, Fig 9B; *P* < 0.001, Figs 9C and S6A–S6C and S14 and S15 Tables). Additionally, higher scores showed a significant positive correlation with LNM progression in individual samples from 214 patients with OSCC (*P* = 0.03, 95% CI, 0.04-0.61, Fig 9D; *P* = 0.03, Spearman correlation = 0.24, Fig 9E). Furthermore, the scores were significantly higher in LNM-positive samples in the analysis of OSCC cells from 17 OSCC samples (*P* < 0.001, 95% CI, 0.76-1.45, Fig 9F) and in samples from the validation spatial cohort of 12 OSCC samples (*P* = 0.03, 95% CI, 0.06-1.35, Fig 9G).

We then validated the prognostic significance of the 23-gene signature in 2 independent datasets (data sourced from the studies GSE41613 and GSE42743 [33]). In the GSE41613 data, the 23-gene signature score was significantly prognostic for both OS (*P* = 0.002; Fig 9H) and DSS (*P* < 0.001; Figs 9I and S6D–S6F and S15 Table), whereas for the GSE42743 data, the score showed a trend of predicting worse OS (*P* = 0.326; Fig 9J) and was prognostic for DSS (*P* = 0.021; Figs 9K and S6G–S6H and S15 Table). To address the limitations arising from facility differences and

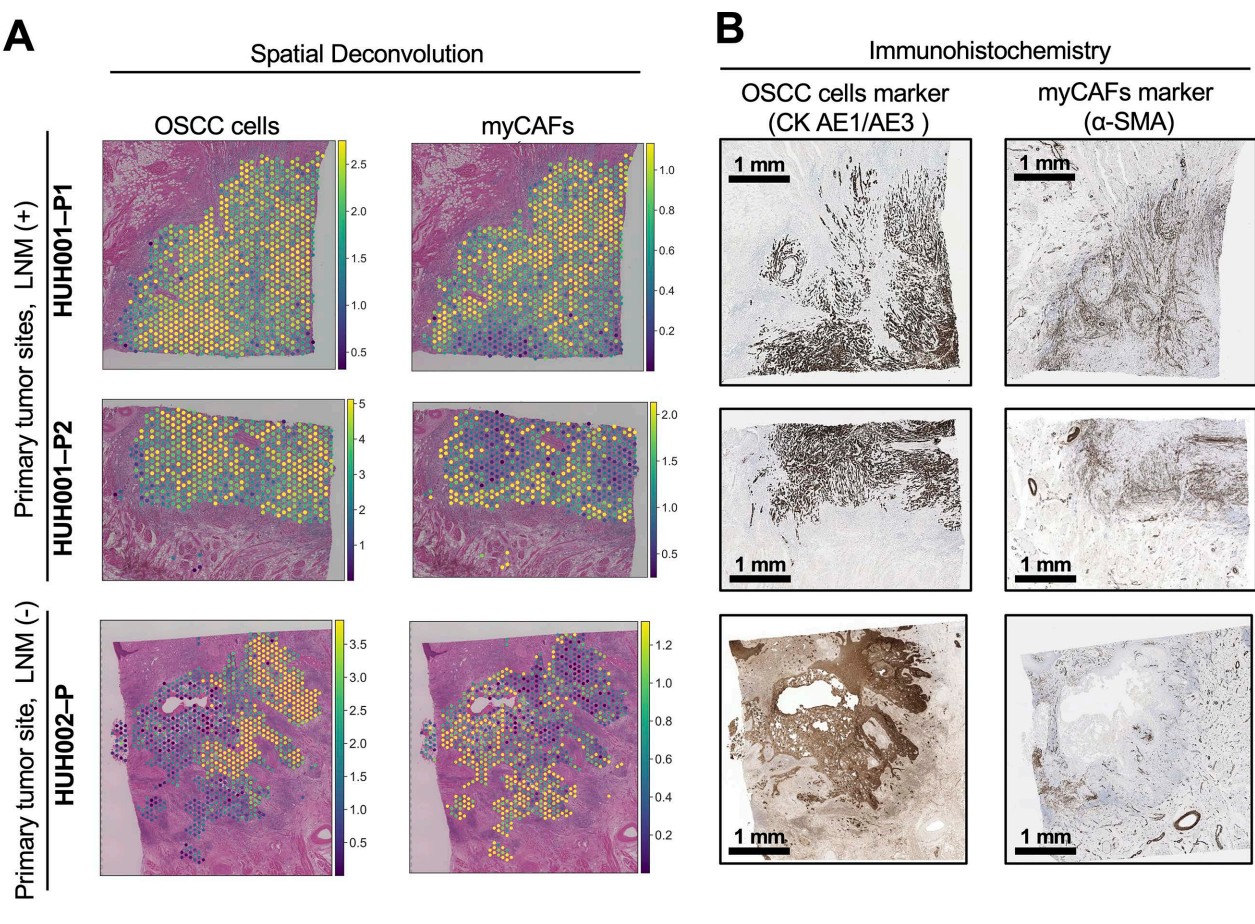

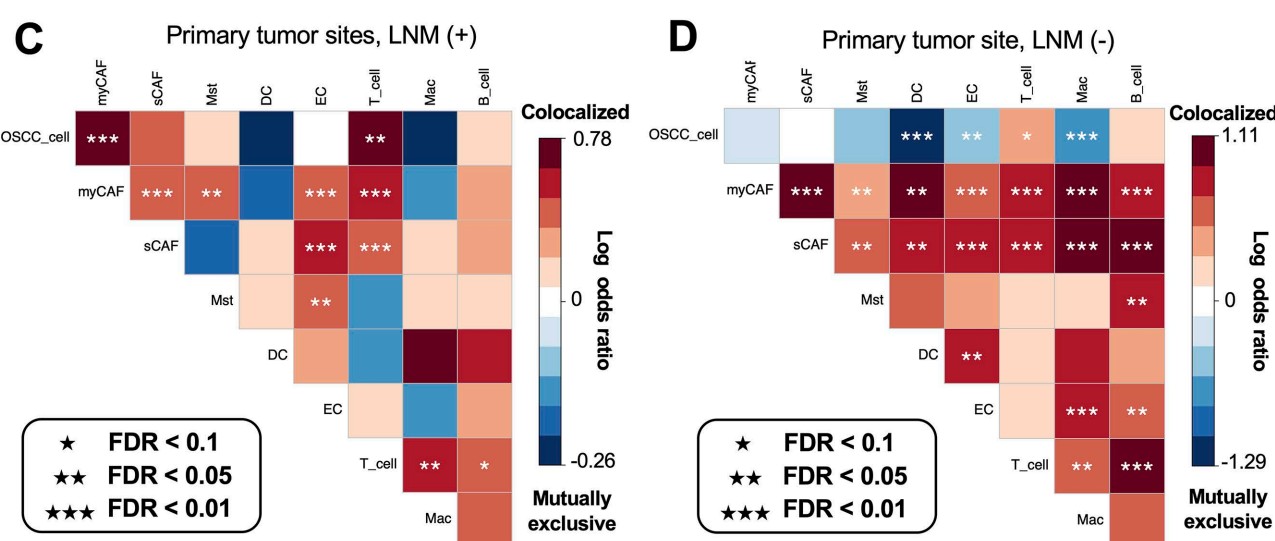

**Fig 6. Spatial mapping and colocalization analyses in primary tumor tissues with or without LNM, as well as IHC confirmation.** (A) Single-cell spatial localization in 2 lymph node metastasis (LNM)-positive samples (HUH001-P1 and HUH001-P2) and a LNM-negative sample (HUH002-P) were analyzed on the basis of the marker genes identified via spatial RNA sequencing. The Tangram method was used to map the predicted annotations of

oral squamous cell carcinoma (OSCC) cells and myofibroblastic cancer-associated fibroblasts (myCAFs) onto Visium Spatial Gene Expression data, and the lower resolution of spatial transcriptomic profiles was addressed at the spot level. The legend shows the estimated cell numbers. See S4A–S4E Fig for more details. (B) The same formalin-fixed, paraffin-embedded (FFPE) samples used for the Visium Spatial Gene Expression analysis underwent immunohistochemical (IHC) staining with cytokeratin/keratin (CK) AE1/AE3 (a marker for OSCC cells), and α-smooth muscle actin (α-SMA, a marker for myCAFs). A 1-mm scale bar is provided for reference. (C, D) Log odds ratios display the spatial relationships among the 8 cell types predicted within the OSCC microenvironment. Spatial colocalization is represented in red, and mutual exclusivity is represented in blue. The odds ratio (OR) was determined using a 2×2 contingency table based on the colocalization counts of cell pairs. The Haldane-Anscombe correction was applied for zero counts. Initially, 10 cell types were identified within the OSCC microenvironment, but those cell types appearing in less than 4% of the total spots were excluded to ensure the robustness and reliability of the statistical analysis by mitigating the influence of sparse data. The level of statistical significance was determined by the false discovery rate (FDR). (C) Primary tumor tissues with LNM (HUH001-P1 and HUH001-P2). (D) Primary tumor tissue without LNM (HUH002-P).

the constrained sample size, we merged the 2 microarray datasets. After harmonizing for batch effects using pyComBat [34,35] and standardizing the combined expression data with a Z-score, we consistently found significant associations between the 23-gene signature and both OS ($P$ = 0.002; Fig 9L) and DSS ($P$ < 0.001; Figs 9M and S6I–S6L and S15 Table). These results suggest that the 23-gene signature that was extracted from the metastatic cluster in the spatial transcriptomic analysis can be used clinically to predict LNM status and poor OS in patients with OSCC.

## Discussion

Using publicly available multiomics data and original spatial transcriptomic data, we studied the role of myCAFs in promoting LNM in OSCC. For analyses using the HNSCC TCGA data, which included various head and neck sub-sites, we applied strict inclusion criteria to define a homogeneous OSCC cohort. We focused on cases annotated as OSCC by site and confirmed their HPV-negative status. While HPV is a critical prognostic factor in HNSCC, particularly in oropharyngeal tumors [36], its prevalence and prognostic impact are considerably lower (<10%) and limited in OSCC [37,38]. Accurately classifying cases and avoiding confounding is challenging in HNSCC cohorts due to anatomical ambiguity (e.g., in tumors of the tongue base or tonsil region) and distinct HPV-positive tumor features [39–41]. Therefore, filtering by both the specific oral cavity site annotation and HPV negativity was crucial. This created a homogeneous, HPV-negative OSCC subtype, including cases from relevant boundary regions whose molecular profiles aligned with those of primary oral cavity tumors, and thereby enhanced the study's validity. Analysis of this cohort revealed several initial insights. We found no significant association between key driver mutations and LNM status, suggesting that genetic events alone may not fully explain the potential for LNM. Instead, bulk RNA-seq analysis highlighted a muscle-contraction transcriptional program enriched in LNM-positive samples, indicating increased stromal activation. We also found a quantitative and functional activation of myCAFs in the LNM-positive microenvironment. In LNM-positive samples, myCAFs spatially colocalized with OSCC cells within the ITF, indicating their active role in promoting LNM. The myCAFs also sent ECM-related signals to OSCC cells, which activated key stemness-related genes in the OSCC cells through ECM-related receptors such as *CD44*. Furthermore, by analyzing the genes specifically expressed in the ITF, where myCAFs and OSCCs colocalize and where metastasis arises, we identified a 23-gene signature that was significantly associated with LNM status and poor OS in patients with OSCC.

Although an increased presence of myCAFs in the TME has been associated with unfavorable outcomes in many solid cancers [12–14,42–44], the role of myCAFs varies depending on the type and the local environment (stromal niche). This difference in environment changes what myCAFs do. For instance, myCAFs can act as tumor suppressors in pancreatic ductal adenocarcinoma (PDAC) [45,46]. PDAC has a very dense stroma with myCAFs that deposit substantial amounts of collagen. This dense stroma forms a barrier that slows down tumor growth and invasion. Studies have shown that removing these myCAFs or reducing this fibrosis makes PDAC grow faster and more aggressively, highlighting myCAFs' protective role in this restrictive niche [47–50]. In contrast, OSCC usually has a less dense, more open stroma. In the OSCC microenvironment, myCAFs gather at the ITF and help OSCC cells invade and spread [12,14]. In our study, we found that

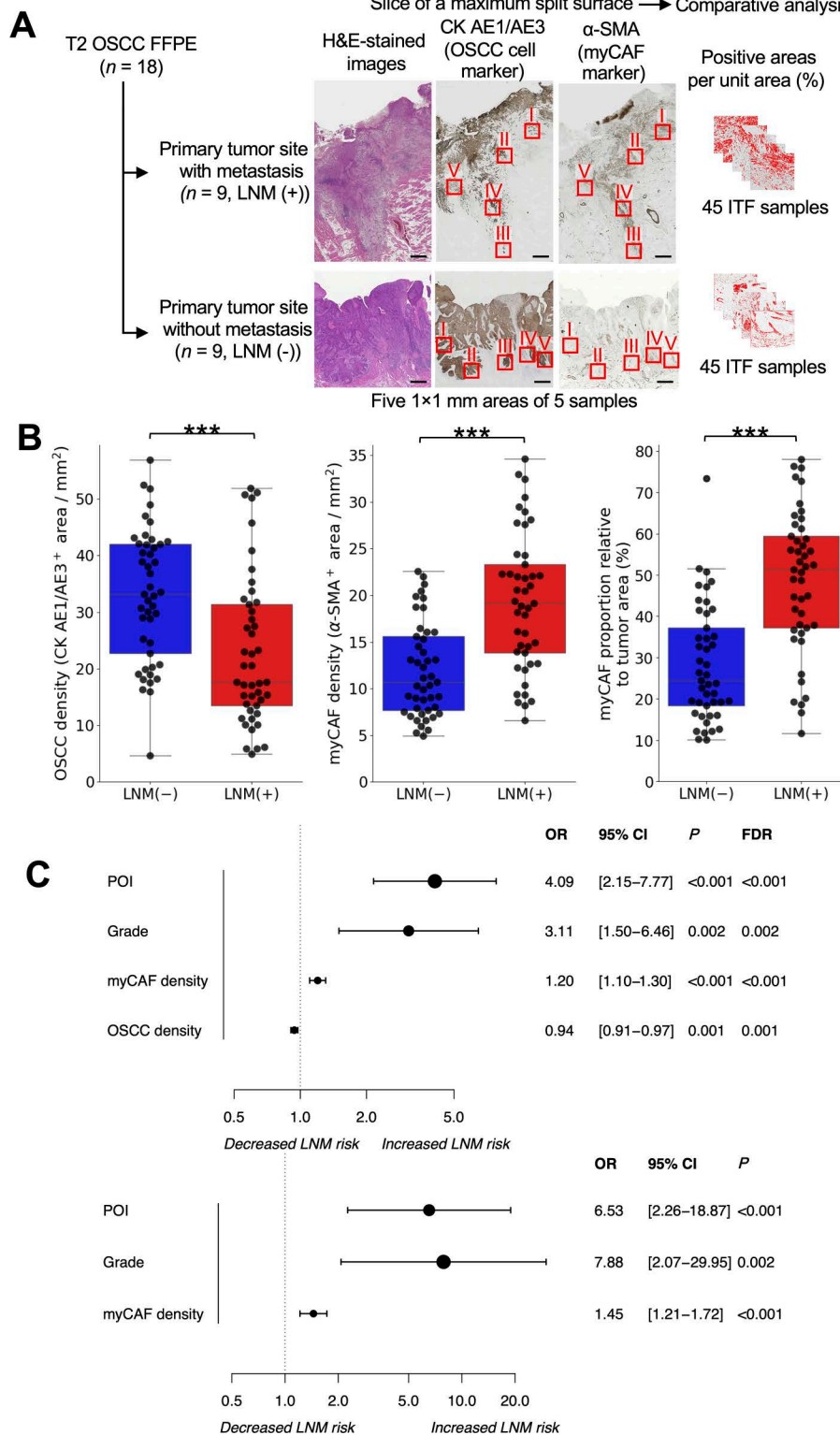

**Fig 7. IHC analysis of the ITFs in 18 OSCC patients by LNM status.** (A) Representation of the IHC analysis conducted on 90 invasive tumor front (ITF) samples. These samples were obtained from 18 patients with OSCC (patients HUH003 to HUH020), with an equal division between patients with and without lymph node metastasis (LNM; see also the IHC analysis of 90 ITF samples from T2 OSCC patients subsection in the Methods and S4

Table). (B) Boxplots illustrate the proportion of IHC-positive regions per unit area across 90 ITF samples, distinguished between those from patients with LNM (represented in red) and those from patients without LNM (in blue). The metrics are the absolute OSCC density (CK AE1/AE3-positive area per square millimeter), the absolute myCAF density (α-SMA-positive area per square millimeter), and the relative myCAF proportion (α-SMA-positive area relative to the sum of the α-SMA- and CK AE1/AE3-positive areas). Each ITF sample is denoted by a black circle. The center line represents the median, the box borders represent the interquartile range (IQR), and the whiskers indicate ± 1.5 × IQR. Statistical significance was assessed using a 2-sided Mann-Whitney $U$ test: *$P < 0.05$; ***$P < 0.001$. (C) Forest plots depict the associations between spatial metrics (OSCC density, myCAF density), pathological differentiation grades, and patterns of invasion (POI) and LNM in OSCC samples. Circles represent odds ratios (ORs), and horizontal bars display 95% CIs. ORs were determined by univariate and multivariate logistic regression analyses. Statistical significance is indicated by a 95% CI that does not cross 1.0.

increased numbers and functionality of myCAFs are associated with an LNM-positive microenvironment. The myCAFs upregulated ECM-related genes, including those in the collagen gene family, in LNM-positive samples. Concurrently, OSCC cells upregulated ECM-associated receptors, specifically *ITGAs*, *ITGBs*, *SDCs*, and *CD44*. This finding aligns with a previous report's finding that COL1A1 produced by fibroblasts interacts with CD44 in cancer cells, facilitating tumor progression in both HPV-positive and HPV-negative HNSCC [51]. Additionally, through spatial transcriptomic and IHC analyses, we determined that the spatial interactions between OSCC cells and myCAFs occurred within the ITF. These findings show that myCAF function strongly depends on the specific stromal niche. Although myCAFs can stop growth in dense PDAC tumors, our findings provide compelling evidence that myCAFs facilitate tumor progression in the OSCC microenvironment. Understanding these differences based on the TME is important for designing future treatments.

Beyond providing insights into the myCAF/OSCC crosstalk facilitating LNM, our 23-gene signature offers prognostic value. While gene expression profiling is not currently standard for OSCC prognosis, our findings suggest that this 23-gene signature, reflecting myCAF activity at the ITF, has the potential to be developed into a clinically relevant gene transcription assay. Similar to the Oncotype DX Breast Recurrence Score test, which uses 21 genes to assess recurrence risk and predict chemotherapy benefit in early-stage breast cancer, this assay could significantly improve risk stratification and clinical management for patients with OSCC. Furthermore, individual components of this signature highlight potential therapeutic avenues (S13 Table). For instance, unfavorable signature genes such as secreted frizzled related protein 2 (*SFRP2*, a Wnt modulator with antibodies in early trials), cellular communication network factor 2 (*CCN2* [*CTGF*], targeted by pamrevlumab), mannose receptor C-type 2 (*MRC2* [*Endo180*], a target for Endo180-ADC platforms), oxidized low density lipoprotein receptor 1 (*OLR1* [*LOX-1*], a target for depleting immunosuppressive cells), and pim-3 proto-oncogene, serine/threonine kinase (*PIM3*, a target for PIM inhibitors like AZD1208) are already under investigation. While the clinical use of this 23-gene signature as a comprehensive assay and the precise roles of its constituent genes within the OSCC microenvironment require rigorous validation, the development of these approaches could provide valuable prognostic tools and novel therapeutic targets in patients with OSCC.

A limitation of this study was the small sample size in the spatial transcriptomic analysis. To overcome this, we validated our findings regarding the spatial colocalization of myCAFs and OSCC cells using 12 additional, publicly available spatial transcriptomic OSCC samples and 90 additional IHC samples (see S4 and S7 Tables for details). These analyses confirmed a significant association between increased myCAF density at the ITF and LNM. Building on this finding, a multivariable analysis of these 90 IHC samples confirmed myCAF density as an independent LNM predictor, even when controlling for pathological grade and POI. Since myCAF density itself did not significantly correlate with the pathological grade and POI, it offers distinct information for predicting LNM beyond these established clinicopathological parameters. Supportive *in vitro* data from other studies also provided mechanistic support for our observed ECM-CD44 axis. Additionally, the prognostic significance of the 23-gene signature that we extracted from the spatial analysis was validated in separate clinical TCGA cohorts and 2 other microarray datasets. We believe that these efforts successfully addressed the limitations regarding the small sample size in the spatial transcriptomic analysis.

In summary, we have provided evidence that the spatial colocalization and molecular crosstalk through the ECM-CD44 axis that occurs between myCAFs and tumor cells influences LNM in patients with OSCC. In addition, through spatial analysis, we extracted

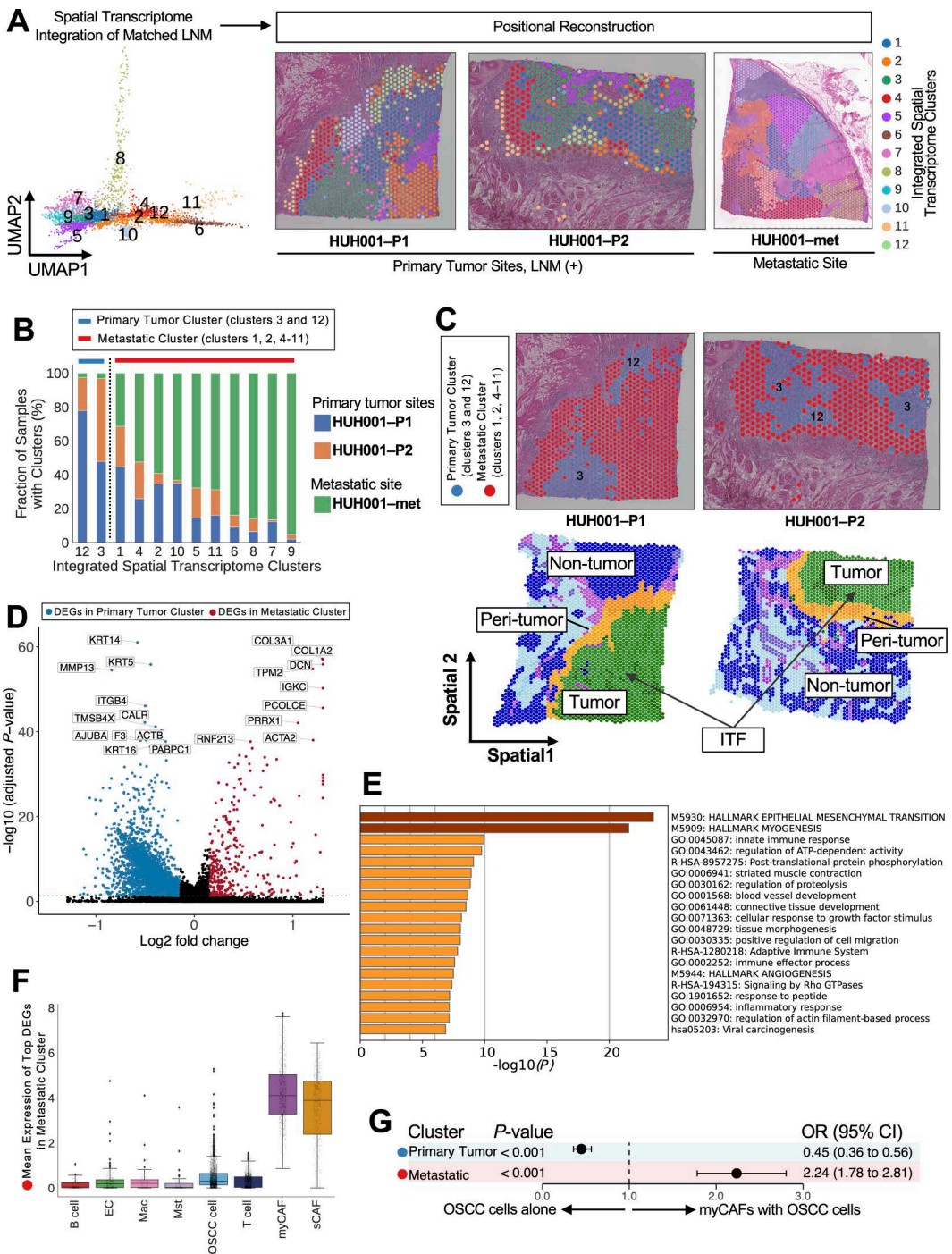

**Fig 8. Integrated spatial transcriptomic analysis of primary tumor and metastatic sites and positional reconstruction.** (A) Uniform manifold approximation and projection (UMAP) visualization shows integrated spatial transcriptome data from 3,637 spots from 2 primary tumor sites (HUH001-P1 and HUH001-P2) and a metastatic site (HUH001-met) from patient HUH001, who had lymph node metastasis (LNM). These samples come from the same patient, patient HUH001, and represent matched primary and metastatic sites. The visualization identifies unsupervised clusters, and 12 clusters are mapped back to their original spatial coordinates. In this visualization, each data point corresponds to a spatial spot. (B) Stacked bar graph shows the samples' integrated distribution in spatial transcriptome clusters, with HUH001-P1 in blue, HUH001-P2 in orange, and HUH001-met in green. The color coding at the top of the bars displays the clusters in the primary tumor sites in blue (clusters 3 and 12) and the clusters in the metastatic primary tumor sites in red (clusters 1, 2, and 4-11). (C) Spatial visualization showcases the primary tumor lusters (clusters 3 and 12) in blue and the

metastatic clusters (clusters 1, 2, 4-11) in red. These clusters are mapped onto the spatial coordinates of samples from HUH001-P1 and HUH001-P2. For information regarding pathologic annotation, see the note in Fig 5B. (D) Volcano plot shows 298 upregulated differentially expressed genes (DEGs) in the metastatic cluster, shown in red. The DEGs in the primary tumor cluster are shown in blue. These DEGs were identified using Scanpy with a 2-sided Mann-Whitney $U$ test (false discovery rate [FDR] < 0.05 and $\log_2$ fold change > 0.15). The top 20 DEGs are labeled. (E) Heatmap displays the most enriched pathways from the Metascape analysis of 298 upregulated DEGs in the metastatic cluster. M5930: Hallmark epithelial mesenchymal transition was the top enriched pathway. (F) Box plots illustrate the average expression of the top 9 DEGs (*COL1A2*, *COL3A1*, *DCN*, *TPM2*, *IGKC*, *PCOLCE*, *PRRX1*, *ACTA2,* and *RNF213*) in 8 distinct cell types in the metastatic cluster, as determined using single-cell RNA sequencing data with the exclusion of cell types representing less than 1% of the total cell population. The black circles indicate individual cells, the center lines represent the median, the box borders represent the interquartile ranges (IQRs), and the whiskers represent ± 1.5 × IQR. (G) Forest plot shows the association between the spatial coordinates of clusters and colocalization with oral squamous cell carcinoma (OSCC) and myofibroblastic cancer-associated fibroblasts (myCAFs) in primary tumor tissues with LNM (HUH001-P1 and HUH001-P2). Circles represent odds ratios (ORs), and bars display the 95% CI.

a 23-gene signature that predicts LNM status and poor prognosis in these patients. These findings provide novel insights into the role of myCAFs in OSCC progression and clinically useful biomarkers to improve prognostication in patients with OSCC.

## Methods

### Samples, ethics, and consent for publication

This study was conducted in accordance with the principles of the Declaration of Helsinki. Clinical specimens from patients with untreated primary OSCC were used for the spatial gene expression analysis. Approval was obtained from the Human Research Ethics Committee of the Hirosaki University Graduate School of Medicine (approval # 2021-081, jRCT1020210053). Written informed consent was obtained from all participants, and consent included the use of all de-identified patient data for publication. Participants were not compensated. Four FFPE samples from 2 patients with OSCC (patients HUH001 and HUH002) were analyzed using Visium. The tissues from patient HUH001, who had LNM, consisted of 2 samples from primary sites (samples HUH001-P1 and HUH001-P2) and 1 sample from a metastatic site (HUH001-met). The tissue from patient HUH002, who did not have LNM, consisted of 1 sample from the primary site (HUH002-P). Case details are provided in the method details section. In the IHC analysis, we included 18 additional patients with OSCC and stage T2 primary tumors (patients HUH003 to HUH020), 9 of whom had LNM and 9 of whom did not (S4 Table). Permission to conduct the IHC analysis was granted by the Human Research Ethics Committee of the Hirosaki University Graduate School of Medicine (approval # 2022-025), which waived the need to obtain informed consent, given the noninterventional, retrospective design of the IHC analysis.

### Bulk WES and RNA-seq data collection and processing

Data from 502 head and neck squamous cell carcinoma cases were obtained from TCGA's database (TCGA-HNSC) [52]. We excluded 268 cases; in 242 cases, the primary tumor site was not the oral mucosa, and in 26 cases, the presence or absence of LNM was pathologically unclear (S1A Fig). We included 234 OSCC cases for which the primary tumor site was the oral mucosa and a pathological diagnosis of LNM or no LNM had been obtained. Open-access data for clinical characteristics, single-nucleotide variations, and transcriptome profiles were obtained using the gdc-client v1.6.0 from the Genomic Data Commons (GDC) Data Portal (RRID:SCR_014514). We reviewed 234 cases of oral cancer from TCGA (RRID:SCR_003193) according to the bulk OSCC data analysis inclusion criteria, and 201 HPV-negative patients were selected, which included LNM ($n = 115$, TNM stage: N1–N3) and non-LNM ($n = 86$, TNM stage: N0) cases (patient selection is described in the S1 Methods and S1A Fig).

### RNA-seq differential expression analysis

After filtering out low-expression genes according to the standard TCC v1.26 (an acronym for Tag Count Comparison; RRID:SCR_001779) procedure [53], iDEGES/edgeR normalization was performed using the iteration option with the

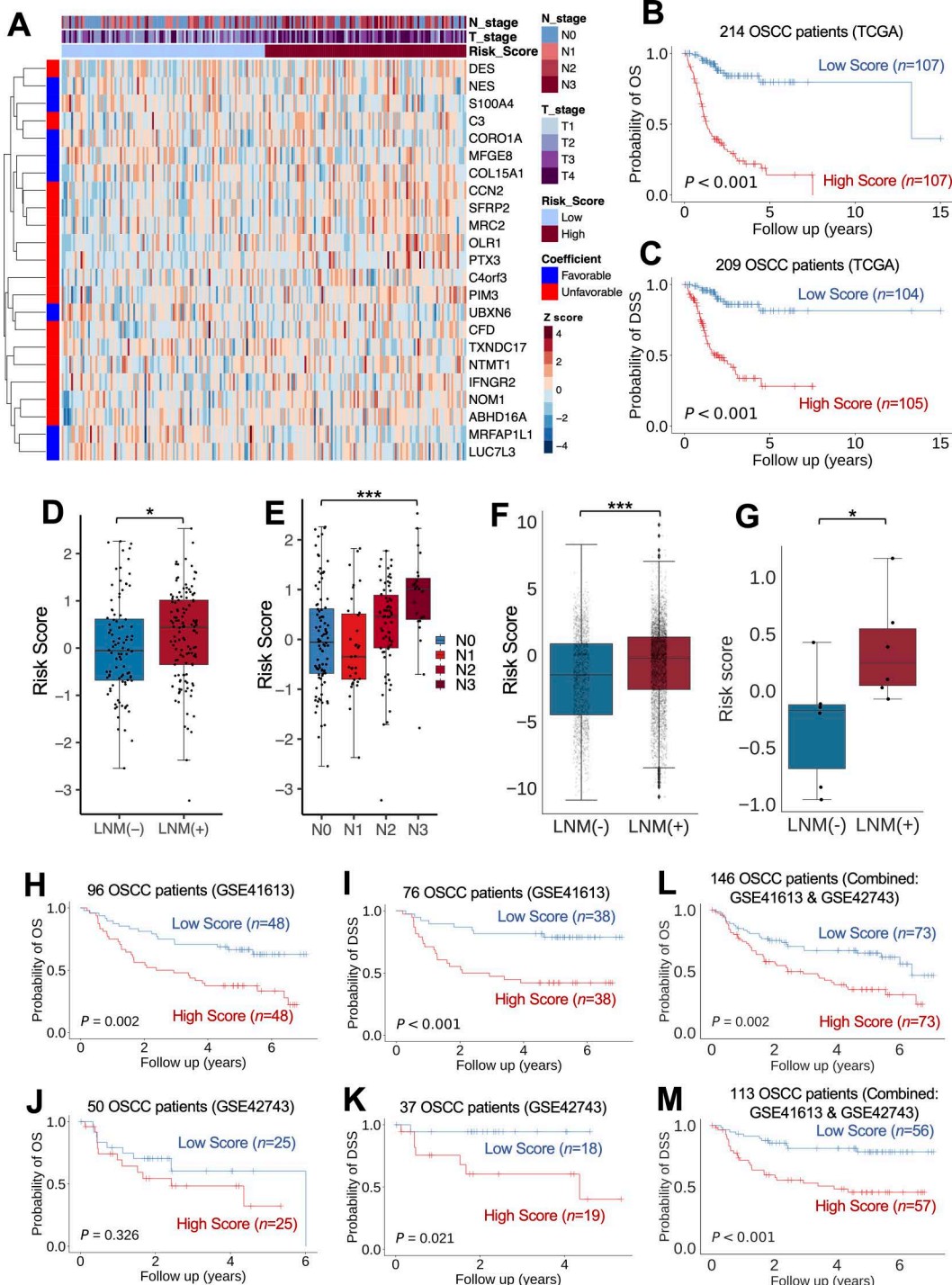

**Fig 9. Validation of spatially resolved 23-gene signature with external transcriptome data.** (A) Heatmap shows the Z-score expression of spatially resolved 23 core genes across 214 patients with oral squamous cell carcinoma (OSCC). Each row represents a gene, and each column represents a patient. The color bar above the heatmap shows clinical information, including lymph node metastasis (LNM) represented as an N stage, tumor size represented as a T stage, and risk scores from the 23-gene signature. On the left, another color bar shows whether each gene coefficient was unfavorable (red) or favorable (blue). (B, C) Kaplan-Meier curves show overall survival (OS) and disease-specific survival (DSS) based on the risk scores from the 23-gene signature and across different platforms. The high-score and low-score groups were determined on the basis of the median risk score. Statistical significance was assessed using log-rank tests. For additional information, please refer to S6A–S6C Fig. (B) OS for 214 patients from The Cancer Genome Atlas (TCGA). (C) DSS for 209 patients from TCGA. (D-G) Box plots depict risk scores from the 23-gene signature by LNM status. Black circles

represent (D, E) 214 individual samples from patients with OSCC and (F) 5,467 OSCC cells across 17 OSCC single-cell RNA sequencing samples. (G) Samples from the validation spatial cohort of 12 OSCC samples. The center line represents the median, the box borders represent the interquartile ranges (IQRs), and the whiskers represent ± 1.5 × IQR. Statistical significance for the group differences was assessed using a 2-sided Mann-Whitney *U* test, and the strength of association was evaluated using the Spearman correlation: *P < 0.05; ***P < 0.001. (H-M) Kaplan-Meier curves show OS and DSS based on the risk scores from the 23-gene signature and across different platforms. The high-score and low-score groups were determined on the basis of the median risk score. Statistical significance was assessed using log-rank tests. For additional information, please refer to S6E–S6N Fig. (H) OS for 96 patients from GSE41613, (I) DSS for 76 patients from GSE41613, (J) OS for 50 patients from GSE42743, (K) DSS for 37 patients from GSE42743, (L) OS for 146 patients from GSE41613 and GSE42743 combined, and (M) DSS for 113 patients from GSE41613 and GSE42743 combined.

number of iterations in the normalization pipeline set to 3. DEGs were detected internally using the edgeR v3.28.1 exact test [54–56] with a cutoff false discovery rate [FDR] of less than 0.01. An enrichment analysis of the genes with variable expression was performed with Metascape (RRID:SCR_016620, https://metascape.org/) [57] using the default parameters (including an enriched term with at least 3 candidates, a *P*-value ≤ 0.01, and an enrichment factor ≥ 1.5) and the analysis species set to *Homo sapiens*. The Gene Ontology biological process, Kyoto Encyclopedia of Genes and Genomes, and Reactome options were selected.

## RNA-seq deconvolution analysis

For each sample, calculations were performed using the ESTIMATE website (https://bioinformatics.mdanderson.org/public-software/estimate/) [15], which scores the purity of tumor tissue, presence of stromal cells, and level of immune cell infiltration based on RNA-seq data. Deconvolution of bulk RNA-seq was performed using CIBERSORTx (RRID:SCR_016955, https://cibersortx.stanford.edu/) [58,59]. The OSCC single-cell RNA-seq dataset (GSE103322) was obtained from the GEO database. A signature matrix was created using the OSCC single-cell RNA-seq dataset, and the parameters were set as replicates, 5; sampling, 0.5; fraction, 0.75; kappa, 999; q-value, 0.01; number of barcode genes, 300; and max, 500). Cell fractions were created by normalizing the OSCC RNA-seq raw data with transcripts per kilobase million, using the signature matrix and 1,000 permutations for statistical analysis. We used the relative cell fractions (the default parameter) for each cell type in the mixed tissues.

## Single-cell transcriptome data processing

The OSCC scRNA-seq dataset (GSE103322) [7], which contains 5,884 cells from 17 of the 18 cases (no transcriptome data were obtained from 1 patient), was used (S2A Fig). Five patients had no LNM (TNM stage: N0) and 12 had LNM (TNM stage: N1-N3), of whom 5 had both primary and metastatic sites. We organized the sample IDs and cell-type annotations on the basis of the source data annotation [7] using Scanpy v1.7.1 (RRID:SCR_018139) [60] (details are provided in S1 Methods and S3 Table and S2F–S2H Fig).

## Cellular scRNA-seq interaction network analysis

Using CellChat (RRID:SCR_021946) [16], which quantitatively measures networks using CellChatDB [61] data on ligand-receptor complexes, we estimated cell-cell interactions in the TMEs of OSCC samples from the metastatic (*n* = 12) and nonmetastatic (*n* = 5) groups. A signaling pathway refers to a set of ligand-receptor pairs belonging to a specific signaling pathway. Following standard CellChat procedures, we first calculated the probability of communication between each cell type using the computeCommunProb function's default parameters to estimate the communication network. Subsequently, we set the filterCommunication function at min.cells = 10 (the default parameter) and filtered cell-cell communications for cell types with fewer than 10 cells. Finally, we used the computeCommunProb pathway function to estimate intercellular communication at the signaling pathway level.

## Gene dependency analysis

For the gene dependency analysis, we used CRISPRGeneEffect data from the DepMap Public 23Q2 primary dataset, which includes dependencies for 17,931 genes across 1,864 cell lines (available at DepMap Portal, https://depmap.org/portal/download/all/). For this analysis, OSCC cells (*n* = 49) and other cell lines, excluding head and neck squamous cell carcinoma cells (*n* = 1,024), were considered. Adjusted effect sizes for gene dependency were determined using the cdsr-models::lin_associations function. Methodological details can be found on the DepMap Community Forum (https://forum.depmap.org/). Statistical significance was assessed using a 2-sided Mann-Whitney *U* test, and the Benjamini-Hochberg method was applied for *P*-value correction.

## FFPE sample preparation and hematoxylin and eosin—stained images for spatial transcriptome analysis

For each surgical specimen, samples were fixed in 10% buffered formalin for 12 h to 24 h. To accommodate the specific requirements of the Visium platform (6.5 mm × 6.5 mm or smaller), 2 adjacent FFPE tissue blocks were prepared from the same specimen: 1 designated for routine pathological diagnosis (including hematoxycin and eosin [H&E] and immunohistochemical staining) and 1 specifically dedicated for Visium spatial transcriptome analysis. The FFPE blocks were stored at 4 °C away from direct sunlight. We first evaluated the messenger RNA quality in the samples from the Visium-dedicated block using an RNeasy FFPE Kit (Qiagen, Hilden, Germany). A good predictor of success in establishing libraries using Visium FFPE is the percentage of 200-base RNA fragments (DV200) greater than 50%. All FFPE samples had high RNA quality with DV200s of greater than 80% (S5 Table). Next, a tissue adhesion test was performed on a test section from the Visium-dedicated block to ensure that the dissected tissue was affixed to the imaging slide. Then, Visium-dedicated FFPE blocks were immersed in ice water for 30 min for rehydration. The 5-µm-thick sections were placed on slides (Visium gene expression slides, 10x Genomics, Pleasanton, CA, USA), dried in a thermal cycler (Bio-Rad C-1000 Thermal Cycler, RRID:SCR_019688, Hercules, CA, USA) at 42 °C for 3 h, and placed in a desiccator to dry overnight. The dried slides were further heated in a thermal cycler at 60 °C for 2 h and deparaffinized using xylene (214736, Millipore Sigma, St. Louis, MO, USA) and ethanol (E7023, Millipore Sigma). Subsequently, the slides were washed with Milli-Q water and stained with hematoxylin (3 min; MSH16, Millipore Sigma), Dako Bluing Buffer (1 min; CS70230-2, Agilent Technologies, Santa Clara, CA, USA), and eosin (1 min; HT110116, Millipore Sigma). The slides were cleaned with Milli-Q water in between steps. Following the completion of staining and the addition of coverslips to the slides, images of the H&E–stained sections were acquired using the apochromatic option on an all-in-one BZ-X800 fluorescence microscope (Keyence, Osaka, Japan) at 10 × -20 × magnification (BZ-PA10-20). To minimize batch effects, sections were prepared on the same day by the same individual and using the same reagent lot.

## Spatial RNA-seq using FFPE Visium

Quality checks of the FFPE Visium libraries were performed using the TapeStation High Sensitivity D5000 system (Agilent Technologies). Library concentrations were determined using a KAPA Library Quantification Kit (KAPA Biosystems, Wilmington, MA, USA). A pool of indexed libraries was sequenced on the NextSeq 550 platform using a High Output Kit v2.5 (Illumina, San Diego, CA, USA), which has a read length of 75 × 2 base pairs with paired-ends and covers at least 25,000 reads per spot, according to the manufacturer's information booklet.

## Spatial RNA-seq data processing and pathological annotation

Reads were demultiplexed and mapped to the reference genome using GRCh38 with Space Ranger software v1.3.0 (10x Genomics). Using the Space Ranger software for each sample, we aligned the H&E–stained images, mapped the entire spatial transcriptome of the FFPE samples, and obtained unsupervised clusters based on the spatial transcriptome (the default setting). Additionally, for all the samples, the whole transcriptomes per spot exceeded the expected metrics values

for the sequencing, mapping, and spots, confirming that the quality was sufficient to proceed with the spatial transcriptome analysis (S6 Table). Count matrices and image data were loaded using the dedicated Spatial Transcriptomics Analysis Software Squidpy v1.1.2 [62] or stLearn v0.3.1. Three pathologists (T.Y., S.G., and H.K.) independently annotated spots using H&E–stained image data on a Loupe browser v6.00 (RRID:SCR_018555, 10x Genomics). Based on the pathological annotations on the H&E–stained images, we classified the spots into 3 major categories: tumor region, peritumor region, and nontumor region. Scanpy was used for quality control for each sample region. First, cells with low gene expression and genes expressed in only a few cells were excluded. We then used the p.filter_cells function to filter out cells with unique molecular identifier counts lower than 500 using the setting min_counts = 500 in the tumor region. For the peritumor and nontumor regions, the p.filter_cells function was set to min_counts = 100. Next, we set min_cells = 2 in the pp.filter_genes function to exclude rare genes detected in 2 or fewer cells. Then, using the p.normalize_total function, each cell was normalized to the total count of all genes, and we equalized the total count of each cell following normalization. Owing to their potential to strongly affect normalization, genes with total unique molecular identifier counts greater than max_fraction = 0.05 in at least 1 cell were considered highly expressed and were excluded from the normalization calculation (excluding_highly_expressed = true). We then used the pp.log1p function to convert the data matrix into natural logarithms.

## Spatial ligand–receptor interaction analysis

Spatial interactions in the OSCC samples were estimated using Squidpy, which stores the distances between observations in a spatial graph and quantifies spatial interactions using a ligand-receptor interaction database [61,63] on the basis of spatial neighborhoods and gene expression levels. First, we performed spatial morphological expression normalization using stLearn v0.3.1 [30], which normalizes gene expression using spatial neighborhood information and morphological distance. Following standard procedures, image data were extracted using the pp.tiling function of stLearn for preprocessing and the cnn_base = exception in the pp.extract_featur function. Features were extracted from the image data using Xception [64], a convolutional neural network. A principal component analysis of the spatial gene expression data was performed using the setting svd_solver = arpack in the em.run_pca function of stLearn, with the logarithm-converted data normalized for quality control for each sample region in Scanpy v1.7.1 [60]. Next, the gr.ligrec function, implemented in Squidpy's ligand-receptor analysis algorithm [61], was used with the setting interactions_params = {resources: CellPhoneDB}, and the ligand-receptor complex CellChatDB [61] was used to identify significant interactions. We set the number of test permutations to 1,000 (n_perms = 1,000) and the cutoff threshold for the percentage of cells needed per cluster to 0.01. We used the Benjamini–Hochberg method with α = 0.05 for the significance level of the FDR correction and the setting corr_method = fdr_bh. The ligand-receptor permutation test results were visualized using the pl.ligrec function (FDR < 0.001 and means_range > 0.95) in Squidpy.

## Spatial RNA-seq mapping analysis

We investigated the spatial localization of cells in the ITFs at the single-cell level. We deconvoluted the cells within each spot according to the standard Tangram procedures for mapping single-cell-level annotations to spatial data [24,65]. We deconvoluted the cells within each spot. Using the OSCC scRNA-seq dataset (GSE103322), we detected signature markers in 10 principal cells using the tl.rank_genes_groups function with the Wilcoxon method in Scanpy v1.7.1 [60]. We identified a set of 338 genes common to both the single-cell and the spatial datasets and used the map_cells_to_space function in Tangram (mode = constrained, learning_rate = 0.1) and graphics processing units (GPUs) to map to the single-cell space.

## IHC staining of the corresponding FFPE samples

IHC of the deparaffinized slides was performed using the standard avidin-oxidase complex method with an automated immunostainer (BenchMark XT; Ventana Medical Systems, Tucson, AZ, USA). In brief, deparaffinized slides were treated with Tris- EDTA (TE) buffer (pH 7.8) at 95 °C for 4 min. Additionally, slides were treated with 5% nonfat dry milk at 37 °C

for 15 min to block endogenous peroxides and proteins. The slides were then incubated with primary antibodies for 60 min at room temperature (20 °C-25 °C). Antibodies against the following proteins were used in the indicated dilution ratios: cytokeratin/keratin (CK) AE1/AE3, an OSCC cell marker (Nichirei 412811; 1:1; clone names, AE1 and AE3) and α-SMA, a myCAF marker (Abcam Cat# M0851; RRID:AB_2313736; Dako; 1:100; clone name, 1A4). Images of CK AE1/AE3 and α-SMA were acquired using an all-in-one BZ-X800 fluorescence microscope (Keyence).

## IHC analysis of 90 ITF samples from T2 OSCC patients

To measure the rates of CK AE1/AE3 positivity and α-SMA positivity in ITF samples, we performed an IHC analysis of 90 ITF samples from 18 patients with T2 primary tumors: 9 patients (45 samples) had LNM, and 9 patients (45 samples) did not (S4 Table). FFPE tissue sections representative of tumor cross-sections underwent standard immunostaining procedures. Images of the tumor sections (stained H&E, CK AE1/AE3, and α-SMA) were acquired using an all-in-one BZ-X800 fluorescence microscope (Keyence). We randomly selected 5 ITF areas (each $1.0 \times 1.0$ mm$^2$) from a wide tumor area. The same regions were extracted from the specimens stained with α-SMA. We binarized the images using ImageJ software (RRID: SCR_003070, Java 1.6.0_24 [64 bit]; National Institutes of Health, Bethesda, MD, USA). The binarized images showed either positive or negative immunostaining. We calculated the percentage of CK AE1/AE3–positive cells (OSCC cells) and α-SMA–positive cells (myCAFs) per unit area ($1.0 \times 1.0$ mm$^2$).

## Spatial transcriptome integration analysis

To integrate the spatial transcriptome data, we extracted the tumor regions from the primary tumor sites (samples HUH001-P1 and HUH001-P2) and the cervical LNM site (HUH001-met) from the samples from the patient with cervical lymph node metastasis (HUH001). Next, using scVI [28] (scvi-tools v0.14.5 [66]), we corrected the sample-specific batch effects on the basis of probabilistic models and deep neural networks. As the scvi-tools model requires raw counts of the spatial transcriptome data rather than normalized expression data, we again used Scanpy v1.7.1 [60] to extract data for the respective tumor regions from samples HUH001-P, HUH001-P2, and HUH001-met. We filtered the cellular and genetic data using the setting min_counts=500 in the pp.filter_cells function to remove cells with unique molecular identifier counts lower than 500, and we used the setting min_cells=3 in the pp.filter_genes function to remove genes detected in fewer than 3 cells. The data from samples HUH001-P, HUH001-P2, and HUH001-met were combined using the concatenate function in anndata v0.7.8. Using anndata, we identified 17,943 genes in the combined data, and we selected the top 4,000 highly expressed differential genes using the p.highly_variable_genes function (n_top_genes=4,000; flavor=seurat_v3) in Scanpy. We used the data setup from the anndata function in scvi-tools to correct the batch effect for each sample (batch_key=sample). A model integrating spatial transcriptome data was constructed using the model.SCVI function in scvi-tools. The parameters of the model were as follows: number of nodes per hidden layer, 128 (default setting); dimensionality of the latent space, 30 (n_latent=30); number of hidden layers used for encoders and decoders, 2 (n_layers=2); dropout rate for the neural network, 0.2 (dropout_rate=0.2); and negative binomial distribution (dispersion=gene, gene_likelihood=nb). The trained model can be found on figshare (https://doi.org/10.6084/m9.figshare.20279025.v1). With the model.get_latent_representation and model.get_normalized_expression functions of scvi-tools, the model output was returned to anndata, allowing interoperation with Scanpy. Using the standard procedure for clustering in Scanpy, the neighborhood graph was computed using data integrated with scvi-tools in the pp.neighbors function [67]. Unsupervised clustering was performed at a resolution of 0.5 in the tl.leiden function [68].The tl.umap function [69] was executed with default parameters to embed the neighborhood graph.

## CNV estimation for spatial RNA-seq

The CNV was estimated using inferCNV v1.11.1 (RRID:SCR_021140, the inferCNV of the Trinity CTAT project; https://github.com/broadinstitute/inferCNV) [29]. We created gene and chromosome position files from the bed file probe set (GRCh38) used for mapping. Furthermore, we created sample annotation files from the tumor and nontumor regions on

the basis of the pathological annotations. Additionally, inferCNV calculated the dynamic threshold for noise removal on the basis of the standard deviation of the average residual expression values. The cutoff options for inferCNV were set to 0.1, and genes with an average expression count of less than 0.1 were filtered out. The denoising filter options (default settings) were used to reduce noise (residual expression values of normal cells) while retaining tumor cell expression values.

## LASSO-penalized Cox proportional hazards model

Following a previously described approach [70,71], we employed a LASSO-penalized Cox proportional hazards model for 220 patients with OSCC using the CoxPHFitter function (patient selection is described in the S1 Methods). Parameters for the lifelines v0.27.8 package [72] were set at l1_ratio = 1.0, penalizer = 0.02, and baseline_estimation_ method = 'breslow'(default). We first obtained a curated TCGA-HNSCC survival dataset [31] ($n = 604$), along with other phenotype data ($n = 612$) and HTSeq Counts data in a $\log_2(x + 1)$-transformed format ($n = 546$), from the University of California, Santa Cruz's Xena platform (RRID:SCR_018938, https://xenabrowser.net/datapages/) [32]. We selected the 220 OSCC patients from the GDC sample sheet of bulk RNA-seq analysis. Subsequently, we processed the HTSeq Counts data by first reverting the $\log_2(x + 1)$ transformation and then using transcripts-per-million normalization, re-applying the $\log_2(x + 1)$ transformation, and finally standardizing the Z-scores for each gene to adjust the data for downstream analyses. For the model training, we used survival outcomes—specifically, OS and its duration—and 298 upregulated DEGs identified in clusters of metastatic primary tumor sites from the spatial transcriptome analysis of matched LNM samples. Through 1,000 iterations of 10% leave-out cross-validations, we evaluated the gene frequencies in each iteration and included those genes chosen in over 90% of the iterations. We averaged the coefficients of these genes with an absolute value exceeding $10^{-2}$ cross the 1,000 iterations. From the 298 DEGs, our model selected 23 genes. We resolved risk scores using their averaged coefficients as weighted expression levels.

## Risk score validation

To assess the association between risk scores and prognosis, we conducted survival analyses for OS and DSS using the Kaplan-Meier method and the log-rank test. Initially, the risk score was evaluated in the training cohort from TCGA. Subsequently, to validate its external reliability, we examined the independent OSCC microarray datasets GSE41613 and GSE42743 using survival data from the GEO database. For these datasets, the Affymetrix human genome U133 Plus 2.0 Array probes were used, and 21 genes were detected out of 23. We excluded patients across all datasets on the basis of the following criteria from previous publications [73–75]: perioperative death (within 30 days after surgery), current treatment with neoadjuvant chemoradiotherapy, positive margins, and recurrent lesions. We identified 221 cases of oral cancer in TCGA (RRID:SCR_003193) using the GDC sample sheet of bulk RNA-seq analysis, and 220 patients were selected (patient selection is described in the S1 Methods and S6A Fig). We then preprocessed and removed missing values for survival time and events. We performed a logarithmic transformation and Z-score standardization of TCGA's RNAseq data after transcripts-per-million normalization. Conversely, after we had verified the gene expression distribution, we standardized the publicly available, normalized microarray datasets using Z-scores, enabling cross-platform comparisons. Apart from this, we combined the 2 normalized microarray datasets after harmonizing them for batch effects using pycombat [34,35] and after standardizing them using Z-scores. We then calculated a risk score for each patient using a resolved linear equation and the median risk score as the cutoff of each cohort on the basis of the balanced data and robustness of risk score distribution as was done in previous studies [70,71]. Two distinct Kaplan-Meier survival curves were created. The first was based on the entire observation period. The second used bootstrapping to assess internal validity, representing the average survival curve with a 95% bootstrap CI. Specifically, survival functions were estimated for each of the 10,000 bootstrap samples. To evaluate the uncertainty of the original dataset, each bootstrap sample was divided into high-score and low-score groups based on the median risk score, followed by log-rank tests. We displayed P-values from these bootstrap samples as histograms and contrasted them with the P-values from the original dataset. As each

bootstrap sample's survival function might possess different time points, all time points were acquired, and survival functions were interpolated to align them to common time points. Due to the increased uncertainty regarding survival arising from our examination of the entire observation period, we limited the bootstrap average survival curve to 1 year after the last event to enhance the curve reliability and make the data easier to interpret.

## Statistics and reproducibility

Statistical analyses were performed using R v3.6.2. No statistical method was used to predetermine the sample size for the bulk genome, transcriptome, single-cell transcriptome, and spatial transcriptome analyses, as these were exploratory studies in which the effect sizes of factors related to LNM were not predetermined. The sample size for the IHC analysis was estimated using G*Power v3.1.9.7 (RRID:SCR_013726) [76] (2-sided Mann-Whitney $U$ test; significance level [α] = 0.05; target power [1 − β] = 0.8; allocation ratio N2/N1 = 1.0; effect size [d] = 0.8). Univariate logistic regression analyses were initially performed. Subsequently, multivariable logistic regression was used to determine the independent predictive value of myCAF density for LNM, after adjusting for pathological grade and the POI, in a clinical cohort (refer to S4 Table for patient characteristics and the specific cohort used for this analysis). A multiple logistic regression model was used to analyze the association between each gene mutation and LNM in 201 patients with OSCC. The number of explanatory variables in the model was set to 9 [77], representing the 9 most frequently mutated genes (*TP53, TTN, FAT1, CDKN2A, NOTCH1, PIK3CA, MUC16, SYNE1* and *CASP8*). We performed a post-hoc analysis for the model with a sample size of 201 patients using G*Power (F test; effect size [f²] = 0.15; significance level [α] = 0.05; total sample size = 201; number of predictors = 9), and the power was 0.98. Continuous variables were evaluated for statistical significance using a 2-sided Mann-Whitney $U$ test. For categorical variables, statistical significance was evaluated using a 2-sided Fisher exact test, and odds ratios were calculated to determine the strength and direction of the association. Furthermore, a Spearman correlation was calculated to assess the association between LNM stage progression and cell proportions. In the Spearman correlation analysis, given that the N3 category was comprised of only 2 patients, we added the 20 patients of N2c, a category signifying more severe, bilateral neck LNM, into the N3 category. This adjustment resulted in a total of 22 patients classified as N3. Two-sided *P*-values less than 0.05 were considered significant. The complete *P*-value data can be found in the publicly available code provided. We calculated 95% bootstrap confidence intervals using 10,000 bootstrap resamples and the 2.5th and 97.5th percentiles. Multiple testing corrections were performed using the Benjamini–Hochberg/FDR method.

## Plots and graphs

We primarily used R v3.6.2 or Python v3.7.12 to generate Figs 1A and 5A were crafted with BioRender.com. The oncoplot (Fig 2A) was constructed using the maftools [78] in R. The volcano plots (Figs 2C and 8D) were constructed using the ggplot2 (RRID: SCR_014601) v3.3.5 in R. The forest plots presented in (e.g., Figs 2B and 8G) and (e.g., Fig 7C) were created using Forester v0.3.0 and forplo v0.2.0, respectively. The heat maps (Figs 2D and 8E) were developed using Metascape. Figs 3A–3C and 8A were created using Scanpy v1.8.2 [60] in Python, and the uniform manifold approximation and projection clustering was executed using the default parameters in leidenalg v0.8.8 [68,79]. Some box-and-whisker diagrams (Figs 3E, 3G, 9D and 9E) were created using ggplot2; the remainder (Figs 3D, 7B, 8F, 9F and 9G) and bar plot (Fig 4A) were created with Matplotlib v3.2.2 (RRID:SCR_008624) and Seaborn v0.11.2 in Python. CellChat v1.4.0 [16] was used to visualize the cell-cell interactions in Fig 4B and 4C. The spatial plots (Figs 5B, 6A, 8A and 8C) were formed using Scanpy [60] and Squidpy v1.1.2 [62] in Python. The pairwise and stacked bar graphs (Figs 3F, 6C, 6D and 8B) were created using ggplot2. The Kaplan-Meier curves (Fig 9B, 9C and 9H–9M) were drawn using KaplanMeierFitter and lifelines v0.27.8 [72], matplotlib, and seaborn. For additional data processing tasks, we used R packages (tidyverse v1.3.1, tidyr v1.1.3, tibble v3.1.4, dplyrr v1.0.7, purrr v0.3.4, and stringr v1.4.0) and Python modules (numpy v1.21.5, pandas v1.3.5, anndata v0.7.8, stLearn v0.3.1, and scvi-tools v0.14.5 [66]).

## Supporting information

**S1 Methods.**
(PDF)

**S1 Table. Clinical characteristics of 201 patients with OSCC (related to** Figs 2 **and** S1**).** The percentages in each category are based on the total number of cases with sufficient information. The significance of continuous variables was evaluated using the two-sided Mann–Whitney U test, while the two-sided Fisher exact test was used for categorical variables. Abbreviations: IQR, interquartile range; LNM, lymph node metastasis; OSCC, oral squamous cell carcinoma.
(PDF)

**S2 Table. Multiple logistic regression for mutated genes and LNM in patients with OSCC (related to** Fig 2B**).** *P*-values lower than 0.05 were considered to indicate statistical significance. Abbreviations: CI, confidence interval; LNM, lymph node metastasis; OSCC, oral squamous cell carcinoma; OR, odds ratio; SE, standard error of the coefficient.
(PDF)

**S3 Table. Characteristics of 17 patients with OSCC derived from single-cell transcriptome data (related to** Figs 3 **and** S2**).** All patient data including Destination, age, sex, pathologic T stage, pathologic N stage, tissue origin, primary tumor site, and grade were obtained from a previous study [7]. Abbreviation: OSCC, oral squamous cell carcinoma.
(PDF)

**S4 Table. Characteristics of 20 patients with OSCC from a clinical cohort (related to** Figs 5–9 **and** S3–S6**).** This table includes all 20 patients in the clinical cohort. Patients HUH001 and HUH002 were used for spatial transcriptomics analysis only. Abbreviations: OSCC, oral squamous cell carcinoma; POI, pattern of invasion.
(PDF)

**S5 Table. RNA quality evaluation with DV200 to predict FFPE assay performance (related to the Methods).** HUH001_met_1 and HUH001_met_2 were prepared from the same FFPE block. HUH001_met_1 was used for all subsequent analyses. Abbreviations: Conc., concentrated; DV200, percentage of RNA fragments longer than or equal to 200 nucleotides; FFPE, formalin-fixed, paraffin-embedded.
(PDF)

**S6 Table. Quality of the sequence and characteristics of the spatial transcriptome analysis (related to the Methods).** Abbreviations: LNM, lymph node metastasis; UMI, Unique Molecular Identifiers.
(PDF)

**S7 Table. Characteristics of 12 samples with OSCC derived from validation spatial transcriptome data (related to** Figs 9G**,** S4L–S4P **and** S5K**).** All data including Sample ID, patient ID, age, pathologic T stage, pathologic N stage, tissue origin, primary tumor site, and grade were obtained from a previous study [25]. Abbreviation: OSCC, oral squamous cell carcinoma.
(PDF)

**S8 Table. Characteristics of HPV negative OSCC and CAF co-culture samples (related to** S4Q–S4S Fig**).** All data including Geo accession, sample ID, cell line, cell type, and treatment condition were obtained from the Gene Expression Omnibus (GEO) dataset GSE279481 [26]. Abbreviations: CAF, cancer-associated fibroblast; HPV, human papilloma virus; OSCC, oral squamous cell carcinoma; SCC, squamous cell carcinoma.
(PDF)

**S9 Table. Characteristics of patient-derived HNSCC and CAF co-culture samples (related to** S4T Fig**).** All data including Geo accession, sample ID, age, sex, cell line, experiment type, tumor location, treatment condition, and stage

were obtained from the Gene Expression Omnibus (GEO) datasets GSE178153/GSE178154 [27]. Abbreviations: CAF, cancer-associated fibroblast; HNSCC, head and neck squamous cell carcinoma.
(PDF)

**S10 Table. *CD44* expression and pseudotime in integrated spatial transcriptome clusters (related to S5H Fig).** Cluster numbers are from the analysis of our Visium spatial transcriptome dataset.
(PDF)

**S11 Table. Molecular characteristics and top DEGs of integrated spatial transcriptome clusters (related to S5H Fig).** Cluster numbers, top differentially expressed genes (DEGs), spatial transcriptome-based cluster names, cluster classifications (metastatic or primary tumor), and relevant references are from the analysis of our Visium spatial transcriptome dataset. Abbreviations: GC, Germinal Center; myCAF, myofibroblastic cancer-associated fibroblast; OSCC, oral squamous cell carcinoma.
(PDF)

**S12 Table. Spatially-resolved 23 core signature genes are ranked by spearman correlation with myCAFs proportion (related to Figs 9 and S6).** Genes labeled "unfavorable" with a positive LASSO weight act as poor-prognosis biomarkers, whereas "favorable" genes with a negative weight act as good-prognosis biomarkers. Abbreviation: myCAF, myofibroblastic cancer-associated fibroblast.
(PDF)

**S13 Table. Biological and clinical relevance of Spatially-resolved 23 signature genes (related to Figs 9 and S6).** Abbreviations: BRCA, breast invasive carcinoma; CRC, colorectal cancer; cSCC, cutaneous squamous cell carcinoma; GBM, glioblastoma multiforme; HCC, hepatocellular carcinoma; HNSCC, head and neck squamous cell carcinoma; ICI, immune checkpoint inhibitor; OSCC, oral squamous cell carcinoma; PDX, patient-derived xenograft; PMN-MDSC, polymorphonuclear myeloid derived suppressor cells.
(PDF)

**S14 Table. Clinical characteristics of 214 patients with OSCC (related to Fig 9).** The percentages in each category are based on the total number of cases with sufficient information. The significance of continuous variables was evaluated using the 2-sided Mann–Whitney *U* test, while the 2-sided Fisher exact test was used for categorical variables. Abbreviations: IQR, interquartile range; OSCC, oral squamous cell carcinoma.
(PDF)

**S15 Table. Comparison of five-year survival rates between low and high score groups across all cohorts (related to Fig 9).** Abbreviations: DSS, disease-specific survival; OS, overall survival; TCGA, The Cancer Genome Atlas.
(PDF)

**S1 Fig. Bulk analysis of OSCC patients (related to Fig 2).** (A) Schematic representation of the patient selection process for bulk whole-exome sequencing (WES) and RNA sequencing (RNA-seq) analysis, excluding the larynx, tonsil, hypopharynx, oropharynx, lip, bones, joints, articular cartilage of other and undefined sites, and ill-defined sites in the lip and oral cavity. Specific exclusions included basaloid squamous cell carcinoma samples, human papilloma virus (HPV)-positive samples, and samples from patients with prior malignancies. (B) Clinical characteristics of a set of 201 patients with OSCC. The significance of the continuous variables was evaluated using the 2-sided Mann-Whitney *U* test, and the significance of the categorical variables was evaluated using the 2-sided Fisher exact test. (C) Boxplots denoting the Estimate score, Stromal score and Immune score of 201 patients with OSCC, with LNM (red) and without LNM (blue). Black circles represent individual samples. The center lines represent the medians, the box borders represent the interquartile ranges (IQRs), and the

whiskers represent ± 1.5 × IQRs. Statistical significance ($P < 0.05$) was evaluated using the 2-sided Mann-Whitney *U* test.
(PDF)

**S2 Fig. Single-cell RNA-seq analysis in the OSCC microenvironment (related to** Figs 3 **and** 4**).** (A) Overview of patients included in the single-cell RNA-seq (scRNA-seq) analysis. (B-D) Comparison of cell numbers in lymph node metastasis (LNM) and non-LNM groups based on the oral squamous cell carcinoma (OSCC) single-cell RNA sequencing (scRNA-seq) data. The comparison involved (B) cell types, (C) sample IDs, and (D) the number of cancer cells, fibroblasts, and T cells. (E) Heatmap illustrating the statistically significant pathways enriched in differentially expressed genes (DEGs) between the LNM and non-LNM groups. Colors denote adjusted *P*-values. (F) Uniform manifold approximation and projection (UMAP) visualizations represent 5,884 cells from 17 OSCC samples analyzed through scRNA-seq and show the expression of *CDH1, TGFβ2, VIM, CXCL12, ACTA2, MCAM, MYH11, IL6, MMP2, PDGFRA, FAP*, and *PDPN*. (G) Heatmap displaying the top 5 DEGs for each cell type. The cell types are as follows: B cells, dendritic cells (DCs), endothelial cells (ECs), macrophages (Macs), mast cells (Msts), myocytes (Myos), OSCC (oral squamous cell carcinoma) cells, T cells, myofibroblastic cancer-associated fibroblasts (myCAFs), and secretory/matrix-remodeling cancer-associated fibroblasts (sCAFs). (H) Heatmap depicting the top 15 DEGs between myCAFs and sCAFs. (I) Relative fractions of the 10 cell types within each sample from 201 patients with OSCC were compared across different LNM stages. The fractions were estimated using a CIBERSORTx analysis. The fractions for the OSCC cells, myCAFs, sCAFs, and T cells are also shown in Fig 3G. (J) Scatter plot comparing interaction strengths for each cell type in LNM-positive vs. LNM-negative samples. Refer to Fig 4B. (K) Scatter plot illustrating differential interaction strengths in myCAFs in LNM-positive vs. LNM-negative samples. (L) Circos plot depicting upregulated outgoing signaling from myCAFs to OSCC cells in LNM-positive samples. (M and N) Box plots illustrate the cancer stemness score [21] and expression-based telomerase enzymatic activity detection (EXTEND) score [22] for OSCC cells in LNM-negative samples and LNM-positive samples. These cancer stemness scores were obtained from https://github.com/NNoureen/EXTEND_datacodes/. The y-axis displays the scores. Black circles indicate individual OSCC cells. The center lines represent the medians, the box borders represent the interquartile ranges (IQRs), and the whiskers represent ± 1.5 × IQRs. Statistical significance was assessed using a 2-sided Mann-Whitney *U* test: ***$P < 0.001$. (O) Volcano plot illustrating differential gene essentiality between OSCC and non-HNSCC cell lines. Data were derived from the DepMap Public 23Q2 Primary dataset, encompassing 1073 cell lines. Gene essentiality scores were analyzed using Scanpy with a 2-sided Mann-Whitney *U* test. Genes exhibiting significantly greater essentiality in OSCC cell lines (False Discovery Rate [FDR] < 0.01; adjusted effect size > 0.1) are highlighted in red (n = 574). Conversely, genes with significantly greater essentiality in non-HNSCC cell lines (FDR < 0.01; adjusted effect size > 0.1) are highlighted in blue. Labeled genes include the top 30 essential genes in OSCC (ranked by FDR), *CD44*, and Syndecans (SDCs).
(PDF)

**S3 Fig. Combined histopathological and spatial transcriptome evaluation of primary and metastatic tissues in patients with OSCC (related to** Fig 5**).** (A-B) Formalin-fixed, paraffin-embedded (FFPE) sample sections were btained from the largest cross-sections of tumors. (A) Two primary tissues (HUH001-P1 and HUH001-P2) from patient HUH001 with LNM. (B) One primary tissue from patient HUH002 without LNM (HUH002-P). Refer to Fig 5B. The adjacent sections from the same specimen were used for diagnostic hematoxylin and eosin (H&E)/ immunohistochemical (IHC) images and Visium images, respectively (see also FFPE sample preparation and H&E-stained images for spatial transcriptome analysis in the Methods). (C-F) Spatial transcriptomes for each sample are displayed. On the top left, unsupervised clustering based on spatial transcriptomes are provided. On the top central, histopathological annotations by specialists are provided and detail tissue states and compositions: CA_WD_MD, cancer, well-differentiated/moderately differentiated; CA_MD_WD, cancer, moderately differentiated/well-differentiated; CA_Inv_Les, cancer, invasive lesions; CA_Fib_Lym,

cancer, fibrosis, lymphocytes; CA_PD_Fib, cancer, poorly differentiated fibrosis; CA_PD_Mus, cancer, poorly differentiated muscle; CA_PD, cancer, poorly differentiated; Inf_Fib, inflammation, fibrosis; Inf_Mus, inflammation, muscle; Adi_Tis, adipose tissue; Bld, blood; Con_Tis_Ves, connective tissue, vessels; Epi, epithelium; Gran_Tis, granulation, tissue; LN_Cap, lymph node capillaries; Lym, lymphocytes; Mus, muscle; Neu, neurons; Neu_Tum, neuron tumor; Uncat, uncategorised; and Ves, vessels. These histopathological annotations are broadly characterized into shades of green, representing tumor regions; shades of yellow, indicating peritumor regions; and shades of blue, indicating nontumor regions. On the top right, broad histopathological categorizations for tumor (green), peritumor (yellow), and nontumor (blue) regions are exhibited. The upper right panel displays the total unique molecular identifiers (UMIs), and lower right panel showcases gene expression by UMI counts. The boxplots represent the UMI count for each category. The center lines represent the medians, the box borders represent the interquartile ranges (IQRs), and the whiskers represent ± 1.5 × IQRs. Black dots denote individual spatial spots. (C) HUH001-P1, (D) HUH001-P2, (E) HUH001-met, and (F) HUH002-P. Histopathological annotations also shown in Fig 5B. (G) Visium spatial transcriptomics image processing pipeline using Space Ranger. The figure shows, from left to right: an original high-resolution H&E-stained tissue image; Space Ranger output illustrating the detected tissue boundary and capture spots; and the tissue image co-registered with Visium slide fiducial markers by Space Ranger for accurate spatial registration.
(PDF)

**S4 Fig. Single-cell-level spatial localization analysis in primary tissues from patients with OSCC (related to Fig 6).** (A-D) Spatial localization at the single-cell level in the primary tumor tissues of patient HUH001 with LNM (HUH001-P1 and HUH001-P2) and of patient HUH002 without LNM (HUH002-P) was performed using spatial RNA-seq (spRNA-seq). The legends indicate the estimated cell numbers. Data for the oral squamous cell carcinoma (OSCC) cells and myofibroblastic cancer-associated fibroblasts (myCAFs) are presented in Fig 6A. (A) HUH001-P1, (B) HUH001-P2, (C) HUH001-met, and (D) HUH002-P. (E) Spatial deconvolution training score and validation plots. (F) Box plots comparing 3 different metrics in OSCC samples from ITF areas. The metrics are absolute OSCC density (CK AE1/AE3-positive area per square millimeter), absolute myCAF density (α-SMA-positive area per square millimeter), and relative myCAF proportion (α-SMA-positive area relative to the sum of α-SMA- and CK AE1/AE3-positive areas). In all box plots, center lines represent medians, boxes represent interquartile ranges (IQRs), and whiskers extend to ± 1.5 × IQRs. Dots represent individual ITF areas. Within this panel, box plots are grouped to show comparisons with the pathological differentiation grade (Grade 1, 2, and 3) in the upper section and with the pattern of invasion (POI) classification in the lower section [80]. Differences across grades and POI classifications for each metric were evaluated using the Kruskal–Wallis and Spearman correlations. (G-I) Spatial gene expression mapping of inferred receptor genes in OSCC cells via scRNAseq, including *SDC1, CD44, SDC4, ITGA3, ITGA5, ITGA6, ITGAV, ITGB1, ITGB4, ITGB8, KRT16,* and *ACTA2*. The color-coded regions represent the expression levels of these genes. (G) HUH001-P1, (H) HUH001-P2, and (I) HUH002-P. (J) Ligand-receptor interaction analysis for HUH001-P1, HUH001-P2, and HUH002-P shows tumor-to-tumor, peritumor, and nontumor regions, highlighting a subset of significant ligand-receptor pairs. Visualization filters were applied to receptor pairs (FDR < 0.001; means_range > 0.95). *P*-values were resolved from permutation testing with 1,000 simulations and were adjusted using the Benjamini-Hochberg method. (K) Validation analysis using the GSE208253 spatial transcriptomics dataset. Four UMAP plots display integrated spatial spots colored by sample ID, LNM status, unsupervised Leiden clusters, and annotated regions (ITFs, intermediate, tumor core). (L) Two box plots compare tumor core signature scores (derived from tumor core DEGs in the original study [25]) and leading edge signature scores (derived from leading edge DEGs in the original study [25]) across the annotated ITF, intermediate, and tumor core regions. For the box plots, the black circles indicate individual splots, the center lines indicate medians, the box borders represent interquartile ranges (IQRs), and the whiskers extend to ±1.5 × IQRs. (M) Spatial deconvolution analysis of cell types in the 12 samples from the validation cohort. Upper panels: spatial deconvolution validation plots for each sample. Lower panels: data for

deconvoluted OSCC cells and myCAFs are presented, showing their spatial localization in each sample. Legends within individual plots indicate estimated cell numbers or proportions per spot. (N) Spatial cell-cell communication in the validation spatial cohort of 12 OSCC samples. Left panel: dot plot of signaling interaction strengths between myCAFs, tumor cores, and ITFs. Y-axis: pathways; X-axis: interacting cell or rejion pairs. Dot size and color denote interaction strength. Right panel: horizontal bar chart of top ligand-receptor (L-R) pairs and their strengths for myCAFs to ITFs communication via the COLLAGEN pathway. Bar color reflects relative L-R strength. (O) Spatial fraction of tumor cores and ITFs in OSCC microenvironments from LNM-positive samples from the validation spatial cohort. The box plot compares tumor cores and ITFs based on the OSCC cell context: OSCC cells alone vs. OSCC cells with myCAFs. Individual points are sample-specific fractions. Each sample is denoted by a black circle. The center line represents the median, the box borders represent the interquartile ranges (IQRs), and the whiskers indicate $\pm 1.5 \times$ IQRs. Statistical significance was assessed using a 2-sided Mann-Whitney $U$ test. The Benjamini-Hochberg FDR was applied for multiple testing corrections. (P) *CD44* expression in the spatial validation cohort. Upper panel: violin plots of spot-level *CD44* expression per sample. Violin elements: width indicates data density, black circles are individual spots, and thick black bars are interquartile ranges (IQRs). Lower panel: violin plots comparing sample-averaged *CD44* expression between LNM-positive and LNM-negative groups. Points are individual sample averages. Violin elements for these averages: width indicates density, thick black bars are IQRs. Statistical significance was assessed using a 2-sided Mann-Whitney $U$ test. (Q) Analysis of ECM-CD44 interactions in co-cultures of HPV-negative tongue squamous cell carcinoma (SCC) cells and cancer-associated fibroblasts (CAFs) using GSE279481 data [26]. Upper panel (left to right): (i) expression levels of key myCAF markers in CAF cell lines (61137, 61162), confirming their myCAF phenotype. (ii) Comparison of *COL1A1* expression in myCAFs (mono-culture vs. co-culture with tongue SCC cells). (iii) Comparison of *CD44* expression in tongue SCC cells (Cal33, Cal27) (mono-culture vs. co-culture with myCAFs); statistical evaluation was performed using the 2-sided Mann-Whitney U test ($P < 0.05$ considered significant). Lower panel: scatter plots showing Pearson correlation (r) between fold changes (FC; co-culture vs. mono-culture) of 17 selected ECM core gene expression in myCAFs and FC of *CD44* expression in tongue SCC cells. Correlations are highlighted with *r* on the plots. (R) Hierarchical Bayesian modeling of the relationship between myCAF-derived ECM gene expression and *CD44* expression in OSCC co-culture data. Upper: diagram illustrating the structure of the hierarchical Bayesian model (Directed Acyclic Graph, DAG) used, with an accompanying table defining model parameters. Lower left panel: diagnostic trace plots for Markov Chain Monte Carlo (MCMC) samples of key model parameters displaying sampling history (left trace) and estimated posterior density (right density) for each parameter shown. Lower middle panel: posterior distribution of the population-level mean slope ($\mu\beta$), representing the overall global association between standardized ECM gene expression in myCAF and standardized *CD44* expression in OSCC cells. The 95% highest density interval (HDI) is indicated, and a vertical dashed line marks a slope of zero. Lower right panel: faceted kernel density estimate (KDE) plots showing the posterior distributions of gene-specific slopes ($\beta$ *gene*) for 17 selected ECM core genes. Each facet displays the density for a single gene, with vertical dashed lines indicating a slope of zero. Heatmap displaying the posterior mean of pair-specific slopes ($\beta$ *pair*) for each of the 17 ECM core genes and each experimental myCAF-OSCC cell line pair. The color scale indicates the magnitude and direction of the slope. Forest plot (right of heatmap) showing the posterior mean (dot) and 95% horizontal error bar (HDI) for the gene-specific slope ($\beta$ *gene*) of each of the 17 ECM core genes, sorted and colored by gene. A vertical dashed line at zero indicates no effect. Bar plot (above heatmap) summarizing overall pair-specific effects. Bars represent the mean of $\beta$ *pair* values (posterior means, averaged across all genes for each specific experimental pair), with error bars indicating the 95% HDI for this aggregated pair effect. Bar colors reflect the magnitude of this mean $\beta$ *pair*. Overlaid jittered dots represent the individual posterior mean $\beta$ *pair* values for each gene within that specific experimental pair, colored by gene. A horizontal dashed line at zero indicates no effect. Y-axis label: "Mean $\beta$ *pair* (95% HDI)". (S) Mediation analysis of the indirect effects of myCAF-derived ECM signaling on *CD44* expression in OSCC cells via the integrin and syndecan pathways (GSE279481 [26]). Top row panel: diagrams illustrating the mediation pathways for (left) integrins (ITGAs/

ITGBs; z1) and (right) syndecans (SDCs; z2). Path coefficients are shown for the effect of ECM gene expression in myCAFs on the mediator ($\beta xz$) and the effect of the mediator on *CD44* expression in OSCC cells ($\beta zy$). Middle row panel: histograms of bootstrap distributions for the calculated indirect effects ($\beta xz \times \beta zy$) mediated via the (left) integrin (z1) and (right) syndecan (z2) pathways. The mean indirect effect and 95% CI are annotated within each plot. Bottom row panel: bar plot summarizing and comparing the mean indirect effects with 95% CIs for the integrin-mediated (z1) and syndecan-mediated (z2) pathways. (For this analysis, the ECM predictor (X) was derived from the first principal component of the fold changes (FCs) of selected collagen gene expressions in myCAFs; mediators (M1: ITGAs/ITGBs, M2: SDCs) were derived from the FCs of their respective gene sets in OSCC cells; and the outcome (Y) was the FC of *CD44* expression in OSCC cells.). (T) Box plots illustrating standardized *CD44* expression in patient-derived head and neck squamous cell carcinoma (HNSCC) cell lines from independent validation datasets (source: GSE178153/GSE178154 [27]), cultured alone or with patient-matched CAFs. Left panel: comparison of standardized CD44 expression in HNSCC cell lines grouped by lymph node metastasis (LNM) status (LNM-negative vs. LNM-positive). Right panel: comparison of standardized *CD44* expression in specified tongue SCC cell lines under mono-culture (control) versus co-culture with CAFs. Symbols denote individual patient-derived tongue SCC cell lines, their LNM status, and tumor stage, as detailed in the inset legend. For both panels, individual data points (representing distinct cell line experiments) are overlaid on the box plots. Box plot center lines indicate medians, box borders represent interquartile ranges (IQRs), and whiskers extend to $\pm 1.5 \times$ IQR. *P*-values were determined using a 2-sided Mann-Whitney *U* test. (PDF)

**S5 Fig. Integrated analysis of primary tumor sites and metastatic sites (related to Fig 8).** (A and B) Uniform manifold approximation and projection (UMAP) visualization demonstrating the integrated spatial transcriptome data from 2 primary tumor sites (HUH001-P1 and HUH001-P2) and 1 metastatic site (HUH001-met) from patient HUH001, who had lymph node metastasis (LNM). Labels on the left represent samples; annotations on the right (CA_MD_WD, cancer, moderately differentiated/well-differentiated CA_PD, cancer, poorly differentiated; CA_PD_Fib, cancer, poorly differentiated fibrosis; CA_PD_Mus, cancer, poorly differentiated muscle; and CA_Fib_Lym, cancer, fibrosis, lymphocytes) pertain to histopathology. (C) Copy number variations within the integrated spatial transcriptome in 2 primary tumor sites (HUH001-P1 and HUH001-P2) and 1 metastatic site (HUH001-met) from patient HUH001 with LNM. (D and E) Stacked bar graphs show the sample distributions in spatial transcriptome clusters, with (D) HUH001-P1 in blue, HUH001-P2 in orange, and HUH001-met in green. (E) Histopathological annotations within the clusters. (F) Metastatic tumor regions exhibited significantly longer pseudotimes in the spatial trajectory analysis, suggesting an advanced developmental state. Box plots compare stLearn-derived pseudotime values between aggregated Primary Tumor Clusters (blue) and Metastatic Clusters (red) from the spatial transcriptomics data (median pseudotimes: metastatic clusters = 0.34 [95% bootstrap CI, 0.33-0.34] vs. primary tumor clusters = 0.19 [95% bootstrap CI, 0.17-0.20]). Black circles represent individual spots. The center lines represent the medians, the box borders represent the interquartile ranges (IQRs), and the whiskers represent $\pm 1.5 \times$ IQRs. Statistical significance was assessed using a 2-sided Mann-Whitney *U* test: \*\*\**P* < 0.001. (G) Spatial trajectory inference reveals potential progression paths between integrated saptial transcriptome clusters using partition-based graph abstraction (PAGA) and diffusion pseudotime (DPT). Left panel: PAGA graph illustrating the connectivity between integrated spatial transcriptome clusters. Nodes are colored according to their cluster identity. The node size is proportional to the number of spots within each cluster, and the thickness of edges represents the confidence of connection between clusters. Right panel: the PAGA graph with nodes colored by DPT values, calculated with the root initialized in primary tumor cluster 3. The arrows on the graph indicate the inferred direction of increasing pseudotime to aid the viewer in visualizing a potential cellular progression trajectory. The color bar displays the pseudotime scale. (H) Characterization of integrated spatial transcriptome Clusters by *CD44* expression, pseudotime, and DEG profiles. Top panel: violin plots illustrating the distribution of *CD44* expression levels across integrated spatial transcriptome clusters. Individual

spatial spots are overlaid. Middle panel: box plots showing the distribution of stLearn-derived pseudotime values for the corresponding clusters. Individual spots are overlaid. The horizontal dashed red line indicates a reference pseudotime, the median pseudotime of the primary tumor cluster 12. Bottom panel: heatmap displaying the mean expression of top5 DEGs across integrated spatial transcriptome clusters. Rows represent clusters, and columns represent genes. The row sidebar annotates clusters as primary tumor (blue) or metastatic clusters (red). (I) Spatial gene expression mapping of the top 8 differentially expressed genes (DEGs), including *COL1A2*, *COL3A1*, *DCN*, *TPM2*, *IGKC*, *PCOLCE*, *PRRX1*, and *ACTA2*, from 298 upregulated DEGs identified in the metastatic cluster. The color-coded regions represent the expression of these genes. (J) Violin plots depict the mean gene expression of the hallmark epithelial-to-mesenchymal transition (EMT) and pan-cancer EMT signatures [81,82] between the primary tumor cluster and the metastatic cluster. The width of each 'violin' corresponds to the density of the data points, with black circles representing individual spatial spots and thick black bars denoting the interquartile ranges (IQRs). Statistical significance was assessed using a 2-sided Mann-Whitney *U* test: ***$P < 0.001$. (K) Spatial heterogeneity of tumor regions and association of metastatic cluster DEG expression with lymph node metastasis (LNM) status in the validation spatial cohort (GSE208253). Left panel: UMAP visualizations of spatial spots. Upper panel UMAP colored by annotated regions: ITFs (red), intermediate (purple), and tumor core (blue). Lower panel UMAP colored by the mean expression score of DEGs characteristic of metastatic clusters (the color scale indicates expression levels). Right panel: Box plots comparing the mean expression of top DEGs from metastatic clusters across the tumor core, intermediate, and ITF regions, further stratified by LNM status (LNM-negative, blue; LNM-positive, red). Black circles indicate individual spatial spots. The center lines represent the medians, the box borders represent the interquartile ranges (IQRs), and the whiskers represent ± 1.5 × IQR. Statistical significance was assessed using a 2-sided Mann-Whitney *U* test: ***$P < 0.001$.
(PDF)

**S6 Fig. External transcriptome data validation for spatially-resolved signature genes (related to** Fig 9**).** (A) Overview of patients used in the survival analysis using The Cancer Genome Atlas (TCGA) data. (B) Bar charts show the distribution of TCGA training group risk scores. Median scores are indicated for 214 patients for overall survival (OS) and 209 patients for disease-specific survival (DSS), respectively. (C) Kaplan-Meier curves, averaged using 10,000 bootstrap resamples with 95% CI, display OS and DSS on the basis of the risk scores from 23 signature genes. Patients were grouped as having high (blue) or low (red) risk scores on the basis of the median scores (OS for 214 TCGA patients and DSS for 209 TCGA patients). The *P*-value distributions from the log-rank test for the bootstrap samples are represented by blue bars, and the *P*-values from the log-rank test of the original data are depicted using red dotted lines. Refer to 9B and 9C. (D) Overview of the cohorts from the microarray datasets GSE41613 and GSE42743, which were used for the validation analysis. (E-H) Bar charts depict the risk score distributions in the GSE41613 and GSE42743 datasets. Median scores are highlighted. Kaplan-Meier curves, produced from 10,000 bootstrap resamples with a 95% CI, display OS and DSS outcomes based on these scores. Patients were grouped as having high (blue) or low (red) risk scores on the basis of the median scores. The *P*-value distributions from the log-rank test for the bootstrap samples are represented by blue bars, and the *P*-values from the log-rank test of the original data are depicted using red dotted lines. (E, F) For GSE41613: the OS was determined using 96 patients, and the DSS was determined using 76 patients. For details using the original data, refer to Fig 9H and 9I. (G, H) For GSE42743: the OS was determined using 50 patients, and the DSS was determined using 37 patients. For details using the original data, refer to Fig 9J and 9K. (I, J) Uniform manifold approximation and projection (UMAP) visualizations from the 2 harmonized microarray datasets (GSE41613 and GSE42743). The left visualization illustrates the data before integration, while the right visualization presents it after batch effect correction using pycombat. (I) OS for 146 patients and (J) DSS for 113 patients. (K) Bar charts depict the risk score distributions from the 2 harmonized microarray datasets (GSE41613 and GSE42743). Median scores are highlighted. The OS was determined using 146 patients, and the DSS was determined using 113 patients. (L) Kaplan-Meier curves, produced from 10,000 bootstrap resamples with a 95% CI, display OS and

DSS outcomes based on risk scores. Patients were grouped as having high (blue) or low (red) risk scores on the basis of the median scores. The OS analysis included 146 patients and the DSS analysis included 113 patients; patients were sourced from 2 harmonized microarray datasets (GSE41613 and GSE42743). The *P*-value distributions from the log-rank test for the bootstrap samples are represented by blue bars, and the *P*-values from the log-rank test of the original data are depicted using red dotted lines. See Fig 9L and 9M for further details using the original data. GDC, Genomic Data Commons; HNSC, head and neck squamous cell carcinoma; OSCC, oral squamous cell carcinoma; RNA-seq, RNA sequencing.
(PDF)

## Acknowledgments

We thank the Scientific Research Facility Center of Hirosaki University Graduate School of Medicine, for granting us use of their TapeStation and the NextSeq550. We gratefully acknowledge the provision of supercomputing resources by the Human Genome Center, the Institute of Medical Science, the University of Tokyo and the NIG supercomputer at Research Organization of Information and Systems National Institute of Genetics. We also acknowledge the support of the High-Performance Computing for research facility at The University of Texas MD Anderson Cancer Center for providing computational resources that have contributed to the research results reported in this paper. We are also grateful to Dr. Meiji Kit-Wan Ma and the technical team at 10x Genomics for their expert assistance. We acknowledge Laura L. Russell at MD Anderson Cancer Center for her scientific editing contributions to the manuscript.

## Author contributions

**Conceptualization:** Ken Furudate, Ryohei Ito, Kosei Kubota, Wataru Kobayashi, Koichi Takahashi.

**Data curation:** Ken Furudate, Tadashi Yoshizawa, Koki Takagi.

**Formal analysis:** Ken Furudate, Tadashi Yoshizawa.

**Funding acquisition:** Ken Furudate, Yuya Sasaki, Ryohei Ito.

**Investigation:** Ken Furudate, Tadashi Yoshizawa, Yuya Sasaki, Kohei Fujikura, Ryohei Ito, Koki Takagi, Kosei Kubota, Koichi Takahashi.

**Methodology:** Ken Furudate, Shuya Kasai, Tadashi Yoshizawa.

**Project administration:** Koichi Takahashi.

**Resources:** Shuya Kasai, Tadashi Yoshizawa, Shintaro Goto, Koki Takagi, Hiroshi Kijima, Kosei Kubota, Ken Itoh, Wataru Kobayashi.

**Supervision:** Yuya Sasaki, Kohei Fujikura, Hiroshi Kijima, Koichi Takahashi.

**Validation:** Ken Furudate, Tadashi Yoshizawa, Shintaro Goto, Tomoyuki Tanaka, Hiroshi Kijima.

**Visualization:** Ken Furudate, Shuya Kasai.

**Writing – original draft:** Ken Furudate.

**Writing – review & editing:** Ken Furudate, Shuya Kasai, Tadashi Yoshizawa, Yuya Sasaki, Kohei Fujikura, Shintaro Goto, Ryohei Ito, Koki Takagi, Tomoyuki Tanaka, Hiroshi Kijima, Kosei Kubota, Ken Itoh, Wataru Kobayashi, Koichi Takahashi.

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
