## [Decision Letter · Decision Letter 0]

14 Apr 2025

PGENETICS-D-25-00169

Spatial colocalization and molecular crosstalk of myofibroblastic CAFs and tumor cells shape lymph node metastasis in oral squamous cell carcinoma

PLOS Genetics

Dear Dr. Furudate,

Thank you for submitting your manuscript to PLOS Genetics. After careful consideration, we feel that it has merit but does not fully meet PLOS Genetics's publication criteria as it currently stands. Therefore, we invite you to submit a revised version of the manuscript that addresses the points raised during the review process.

Please submit your revised manuscript within 60 days Jun 09 2025 11:59PM. If you will need more time than this to complete your revisions, please reply to this message or contact the journal office at plosgenetics@plos.org. Please include the following items when submitting your revised manuscript:

We look forward to receiving your revised manuscript.

Kind regards,

John Prensner

Academic Editor

PLOS Genetics

Marnie Blewitt

Section Editor

PLOS Genetics

Aimée Dudley

Editor-in-Chief

PLOS Genetics

Anne Goriely

Editor-in-Chief

PLOS Genetics

**Journal Requirements:**

At this stage, the following Authors/Authors require contributions: Ken Furudate, Shuya Kasai, Tadashi Yoshizawa, Yuya Sasaki, Kohei Fujikura, Shintaro Goto, Ryohei Ito, Tomoyuki Tanaka, Hiroshi Kijima, Kosei Kubota, Ken Itoh, Wataru Kobayashi, and Koichi Takahashi. Please ensure that the full contributions of each author are acknowledged in the "Add/Edit/Remove Authors" section of our submission form.

The list of CRediT author contributions may be found here: https://journals.plos.org/plosgenetics/s/authorship#loc-author-contributions

https://journals.plos.org/plosgenetics/s/submission-guidelines#loc-parts-of-a-submission

5) We have noticed that you have uploaded Supporting Information files, but you have not included a list of legends. Please add a full list of legends for your Supporting Information files after the references list.

Potential Copyright Issues:

i) Please confirm (a) that you are the photographer of 3B, S3A, and S3B, or (b) provide written permission from the photographer to publish the photo(s) under our CC BY 4.0 license.

ii) Figure 3A. Please confirm whether you drew the images / clip-art within the figure panels by hand. If you did not draw the images, please provide (a) a link to the source of the images or icons and their license / terms of use; or (b) written permission from the copyright holder to publish the images or icons under our CC BY 4.0 license. Alternatively, you may replace the images with open source alternatives. See these open source resources you may use to replace images / clip-art:

7) Thank you for stating "The source code can be found at "https://kenflab.github.io/oscc_metastasis/." We notice that there is a CC BY-NC-ND license on your data. We would encourage you to consider using a license that is no more restrictive than CC BY, in line with PLOS’ recommendation on licensing (http://journals.plos.org/plosone/s/licenses-and-copyright). 

8) Please amend your detailed Financial Disclosure statement. This is published with the article. It must therefore be completed in full sentences and contain the exact wording you wish to be published.

9) Thank you for stating "The curated TCGA survival datasets for HNSCC are available from the University of California, Santa Cruz’s Xena platform (RRID:SCR_018938, "https://xenabrowser.net/datapages)." The provided link reaches a DOI Not Found page. Please amend this to a working link.

**Reviewers' comments:**

Reviewer's Responses to Questions

Reviewer #1: This study provides a comprehensive investigation into the role of myCAFs in promoting lymph node metastasis (LNM) in OSCC. While the findings are compelling, several issues require clarification or expansion:

1.In spatial transcriptomic analysis part, can the author supply the clear HE results (without any label) of every sample to clarify the location of tumor cells and myCAFs? Because in the HUH002-P, the location of OSCC cells seems not very accurate based on the picture (left).

2.Through spatial transcriptomic analysis, the author summarized the difference of myCAFS localization between LNM+sample and LNM-sample. Is the difference also related with the morphological characteristics of SCC nests (such as nest-like, lobulated, etc.) or with pathological grade?

3. The sample size in this study is too small to rule out the influence of the pathological morphology of the sample tissues on the results in figure 4. Although 50 additional IHC samples were used, but only from 10 patients, why not use 50 samples from 50 patients (one sample per patient) considering the different samples from one person have similar or same differentiation?

4.The clinical application potential of the 23 gene markers should be described or discussed in more depth.

Reviewer #2: In the manuscript, the authors tried to identify the molecular mechanisms behind the development of lymph node metastasis and poor prognosis of OSCC using spatial transcriptomic analyses. To support their findings, they also used public data.

The study showed some interesting points in regards to OSCC by spatial transcriptomic analyses from VISIUM, 10x genomics. myCAFs were found spatially colocalized with OSCC cells in the invasive front of the tumor. These CAFs were found to upregulate ECM-related genes and ECM outgoing signaling from myCAFs to OSCC cells in LNM positive samples. Especially the authors identified the 23-gene signature associated with LNM status that predicted poor overall survival in OSCC patients.

Although the research objectives and results are interesting, there are some areas that could be improved to further clarify the findings:

1. It would be interesting to expand the study by identifying a specific tumor cluster associated with metastasis to the lymph nodes. For example, analyzing the developmental trajectories of tumor cells could be a valuable addition.

2. The discussion section should be expanded to reflect all the results obtained in the study. For instance, there is a lack of discussion regarding whole-exome sequencing and bulk transcriptome sequencing data from TCGA project.

Reviewer #3: This study represented by Furudate et al. integrats in-house spatial transcriptomic data, and third party bulk transcriptomic, single-cell RNA-seq data, and dissects the role of myCAFs in OSCC progression. The spatial colocalization of myCAFs and OSCC cells at the invasive tumor front (ITF) and their crosstalk via ECM-related signaling and CD44 stemness pathway offer fresh insights into LNM pathogenesis. The identification of a 23-gene signature with prognostic value is clinically significant. Advanced computational tools, such as CellChat, CIBERSORTx, Tangram, and validation via IHC in 50 additional samples strengthen the conclusions. The survival analyses in TCGA and external cohorts enhance reproducibility. However, several issues require clarification or elaboration to strengthen the manuscript.

Key Concerns:

1. The spatial analysis relies on only 2 patients (1 LNM-positive, 1 LNM-negative). While IHC validation in 50 samples mitigates this, expanding spatial transcriptomics to a larger cohort (at least 3-5 pairs) would improve statistical power and generalizability.

2. The proposed ECM-CD44 axis driving stemness in OSCC cells is compelling but requires functional validation. In vitro co-culture experiments (myCAFs + OSCC cells) or in vivo models like xenografts with myCAF depletion could confirm causality. As the DEGs are derived from LNM different samples, it might be better to establish a LNM model in vitro or in vivo and further validate their findings.

3. The discussion contrasts myCAF tumor-suppressive roles in pancreatic cancer with their pro-metastatic role here. A deeper exploration of tissue-specific CAF heterogeneity (e.g., comparing OSCC vs. PDAC stromal niches) would contextualize these findings.

4. The reliance on HPV-negative TCGA cases is justified but could be explicitly discussed in the context of HPV’s known influence on HNSCC prognosis.

Minor comments:

1. In Figure 1C, ensure all gene names/abbreviations are defined (e.g., SYNE1, LRP1B).

2. In Figure 4F, clarify how α-SMA positivity was quantified, which methodology used like percentage area or cell counts.

3. Address whether myCAF-OSCC interactions are exclusive to the ITF or also occur in central tumor regions.

4. Please define abbreviations at first mention, such as TME in the abstract, scVI in Results, etc.

**Have all data underlying the figures and results presented in the manuscript been provided?**

Reviewer #1: Yes

Reviewer #2: Yes

Reviewer #3: Yes

PLOS authors have the option to publish the peer review history of their article (what does this mean? ). If published, this will include your full peer review and any attached files.

**Do you want your identity to be public for this peer review?** For information about this choice, including consent withdrawal, please see our Privacy Policy .

Reviewer #1: No

Reviewer #2: No

Reviewer #3: No

**Figure resubmission:**
---

## [Decision Letter · Decision Letter 1]

30 Jun 2025

Dear Dr Furudate,

We are pleased to inform you that your manuscript entitled "Spatial colocalization and molecular crosstalk of myofibroblastic CAFs and tumor cells shape lymph node metastasis in oral squamous cell carcinoma" has been editorially accepted for publication in PLOS Genetics. Congratulations!

Yours sincerely,

John Prensner

Academic Editor

PLOS Genetics

Marnie Blewitt

Section Editor

PLOS Genetics

Aimée Dudley

Editor-in-Chief

PLOS Genetics

Anne Goriely

Editor-in-Chief

PLOS Genetics

Comments from the reviewers (if applicable):

Reviewer's Responses to Questions

**Comments to the Authors:**

Reviewer #1: All the revisions are good. There are no more problems.

Reviewer #2: All comments have been resolved. The work is valuable and can significantly contribute to the scientific community in oral cancer.

Reviewer #3: I appreciate the author's responses and think they are suitable for publication.

**Have all data underlying the figures and results presented in the manuscript been provided?**

Reviewer #1: Yes

Reviewer #2: Yes

Reviewer #3: None

PLOS authors have the option to publish the peer review history of their article (what does this mean? ). If published, this will include your full peer review and any attached files.

**Do you want your identity to be public for this peer review?** For information about this choice, including consent withdrawal, please see our Privacy Policy .

Reviewer #1: No

Reviewer #2: No

Reviewer #3: No

**Data Deposition**

http://datadryad.org/submit?journalID=pgenetics&manu=PGENETICS-D-25-00169R1

**Press Queries**

---

## [Editor Report · Acceptance letter]

PGENETICS-D-25-00169R1

Spatial colocalization and molecular crosstalk of myofibroblastic CAFs and tumor cells shape lymph node metastasis in oral squamous cell carcinoma

Dear Dr Furudate,

We are pleased to inform you that your manuscript entitled " 

Spatial colocalization and molecular crosstalk of myofibroblastic CAFs and tumor cells shape lymph node metastasis in oral squamous cell carcinoma" has been formally accepted for publication in PLOS Genetics! Your manuscript is now with our production department and you will be notified of the publication date in due course.

With kind regards,

Anita Estes

PLOS Genetics

On behalf of:
